# RETHINKING CONTENT AND STYLE: EXPLORING BIAS FOR UNSUPERVISED DISENTANGLEMENT

## ABSTRACT

Content and style (C-S) disentanglement intends to decompose the underlying explanatory factors of objects into two independent latent spaces. Aiming for unsupervised disentanglement, we introduce an inductive bias to our formulation by assigning different and independent roles to content and style when approximating the real data distributions. The content embeddings of individual images are forced to share a common distribution. The style embeddings encoding instance-specific features are used to customize the shared distribution. The experiments on several popular datasets demonstrate that our method achieves the state-of-the-art disentanglement compared to other unsupervised approaches and comparable or even better results than supervised methods. Furthermore, as a new application of C-S disentanglement, we propose to generate multi-view images from a single view image for 3D reconstruction.

## 1 INTRODUCTION

The disentanglement task aims to recover the underlying explanatory factors of natural images into different dimensions of latent space, and provide an informative representation for tasks like image translation (Wu et al., 2019b; Kotovenko et al., 2019), domain adaptation (Li et al., 2019; Zou et al., 2020) and geometric attributes extraction (Wu et al., 2019c; Xing et al., 2019), etc.

The previous methods (Kim & Mnih, 2018; Higgins et al., 2017; Burgess et al., 2018a; Kumar et al., 2017) learn disentangled factors by optimizing the total correlation in an unsupervised manner. However, Locatello et al. (2019) prove that unsupervised disentanglement is fundamentally impossible without inductive bias on both model and data.

In this paper, we focus on content and style (C-S) disentanglement, where content and style represent two separate groups of factors. The main novelty of our work is that we assign different roles to the content and style in modeling the image distribution instead of treating the factors equally, which is the inductive bias introduced in our method. Most of the previous C-S disentanglement works (Denton & Birodkar, 2017; Jha et al., 2018; Bouchacourt et al., 2018; Gabbay & Hoshen, 2020) rely on supervision, which is hard to obtain for real data. E.g., Gabbay & Hoshen (2020) leverage group observation to achieve disentanglement by forcing images from the same group to share a common embedding. To our best knowledge, the only exception is Wu et al. (2019c). However, this method forces the content path to learn geometric structure limited by 2D landmarks.

Our definition of content and style is similar to Gabbay & Hoshen (2020), where the content includes the information which can be transferred among groups and style is image-specific information. When group observation is not available, we define content includes the factors shared across the whole dataset, such as pose. Take the human face dataset CelebA (Liu et al., 2015) as an example, the content encodes pose, and style encodes identity, and multi-views of the same identity have the same style embeddings, but different content embeddings, i.e., poses.

Based on the above definitions, we propose a new problem formulation and network architecture by introducing an inductive bias: assigning different and independent roles to content and style when approximating the real data distributions. Specifically, as shown in Figure 1, we force the content embeddings of individual images to share a common distribution, and the style embeddings are used to scale and shift the common distribution to match target image distribution via a generator.

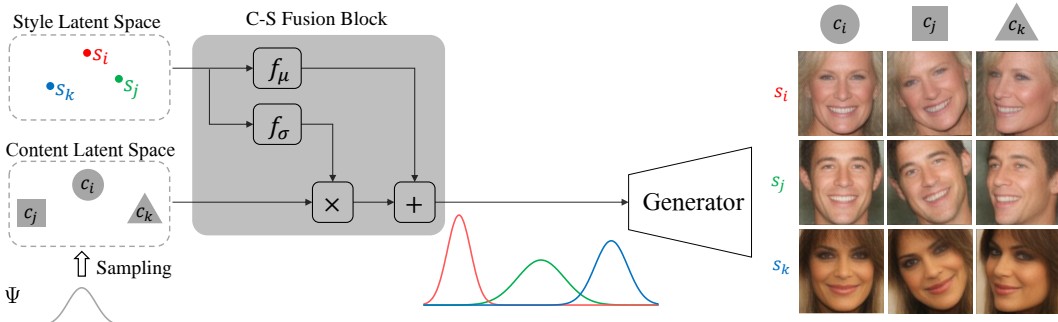

Figure 1: Overview of our framework. $c_i, c_j, c_k$ labelled with different shapes are embeddings sampled from a shared distribution $\Psi$. $s_i, s_j, s_k$ labelled with different colors are embeddings from the style latent space. $f_\sigma$ and $f_\mu$ are two fully-connected layers predicting the statistics parameters to scale and shift $\Psi$ respectively to approximate the target image distributions via a Generator. For each generated image from $3 \times 3$ grid, the content and style embeddings are from the column and row respectively.

We follow Bojanowski et al. (2018) and Gabbay & Hoshen (2020) to apply latent optimization to optimize the embeddings and the parameters of the generator. We also propose to use instance discrimination as a complementary constraint to assist the disentanglement. Please note that we only use the image reconstruction loss as the supervision; no extra labeling is needed. As the content and style perform a different and independent role when modeling the data, they are disentangled to encode the shared and instance-specific features respectively after the optimization.

The contributions of our work are as follows: we achieve unsupervised C-S disentanglement by introducing an inductive bias in our formulation: assign different and independent roles to content and style when modeling the real data distributions. Furthermore, we achieve better C-S disentanglement by leveraging instance discrimination. The experiments on several popular datasets demonstrate that our method achieves the state-of-the-art unsupervised C-S disentanglement and comparable or even better results than supervised methods. Besides, we propose to apply C-S disengagement to a new task: single view 3D reconstruction.

## 2 RELATED WORK

**Unsupervised Disentanglement.** A disentangled representation can be defined as one where individual latent units are sensitive to changes in individual generative factors. There have been a lot of studies on unsupervised disentangled representation learning (Higgins et al., 2017; Burgess et al., 2018a; Kumar et al., 2017; Kim & Mnih, 2018; Chen et al., 2018). These models learn disentangled factors by factorizing aggregated posterior. They can also be used for C-S disentanglement. The learned factors can be divided into two categories; one is content-related, the other is style-related. However, Locatello et al. (2019) proved that unsupervised disentanglement is impossible without introducing inductive bias on both models and data. Therefore, these models are currently unable to obtain a promising disentangled representation. Motivated by Locatello et al. (2019), we revisit and formulate the unsupervised C-S disentanglement problem to introduce inductive bias.

**C-S Disentanglement.** Originated from style transfer, most of the prior works on C-S disentanglement divide latent variables into two spaces relying on supervision. To achieve disentanglement, Mathieu et al. (2016) and Szabó et al. (2018) combine the adversarial constraint and auto-encoders. Meanwhile, VAE (Kingma & Welling, 2014) is used with non-adversarial constraints, such as cycle consistency (Jha et al., 2018) and evidence accumulation (Bouchacourt et al., 2018). Furthermore, latent optimization is shown to be superior to amortized inference (Gabbay & Hoshen, 2020). Unlike the above works, Wu et al. (2019c) propose a variational U-Net with structure learning for disentanglement in an unsupervised manner. However, this method is limited by the learning of 2D landmarks. In our paper, we formulate C-S disentanglement and explore inductive bias for unsupervised disentanglement. Note that style transfer aims at modifying the domain style of an image while preserving its content, and its formulation focuses on the relation between domains (Huang et al., 2018a). Our formulation is defined in a single domain but can be extended to cross-domain, as presented in Appendix G.

# 3 EXPLORING INDUCTIVE BIAS FOR C-S DISENTANGLEMENT

In this section, We first formulate the C-S disentangle problem by exploring the inductive bias and propose a C-S fusion block based on our formulation. We then perform the ablation study to demonstrate how the C-S get disentangled. Finally, our loss functions are presented.

## 3.1 PROBLEM FORMULATION

We parametric the target distribution $P_i(\boldsymbol{x}|\boldsymbol{c})$ as $\hat{P}_{\theta,s_i}(\boldsymbol{x}|\boldsymbol{c})$, where $\theta$ is the parameter of the generator $G_\theta$ that maps embeddings to images, $s_i$ is the style embedding assigned to $I_i$. In our formulation of $\hat{P}$, we assign independent roles to content and style embeddings. $\{c_i\}_{i=1}^N$ are sampled from the distribution of the shared conditional variable $\boldsymbol{c}$, which is denoted as $\Psi$. $\{s_i\}_{i=1}^N$ are the parameters to characterize $\hat{P}$. Thus our inductive bias is introduced into our formulation. $\Psi$ should be close to the ground truth distribution of the dataset, e.g., Gaussian distribution, Uniform distribution, etc.

Our optimization target is to maximize the log-likelihood of $\hat{P}$, and force $\boldsymbol{c}$ to follow the shared distribution $\Psi$ meanwhile:

$$\max_{\theta,c_i,s_i} \sum_{i=1}^N \mathbb{E}_{I_i \sim P_i} \log \hat{P}_{\theta,s_i}(\boldsymbol{x} = I_i | \boldsymbol{c} = c_i),$$

$$s.t. \quad KL(p(\boldsymbol{c})||\Psi) \leq 0,$$

(1)

where $\boldsymbol{c} = c_i$ indicates $c_i$ is the embedding assigned to $I_i$, and $p(\boldsymbol{c})$ denotes the distribution of $\{c_i\}_{i=1}^N$. To solve this problem, we introduce the Lagrange Multiplier as

$$\min_{\theta,c_i,s_i} - \sum_{i=1}^N \mathbb{E}_{I_i \sim P_i} \log \hat{P}_{\theta,s_i}(\boldsymbol{x} = I_i | \boldsymbol{c} = c_i) + \lambda KL(p(\boldsymbol{c})||\Psi).$$

(2)

## 3.2 PROPOSED NETWORK ARCHITECTURE

Here we propose a network architecture to address the formulated problem in Section. 3.1. In particular, we design a C-S fusion block to assign different roles to content and style in modeling real data distribution. As shown in Figure 1, we add this block before the generator to force the input to follow the customized distribution.

Inspired by the observation that mean and variance of features carry the style information (Gatys et al., 2016; Li & Wand, 2016; Li et al., 2017; Huang & Belongie, 2017), we use the style embeddings to provide the statistics to scale and shift the shared distribution $\Psi$ to match the target distribution as

$$z_i = f_\sigma(s_i) \cdot c_i + f_\mu(s_i),$$

(3)

where $f_\sigma$ and $f_\mu$ are two fully connected layers predicting the mean and variance respectively. With this design, Eq. 1 is equivalent to minimize

$$\min_{\theta,c_i,s_i} \sum_{i=1}^N \|I_i - G_\theta(z_i)\| + \lambda KL(p(\boldsymbol{c})||\Psi).$$

(4)

Please refer to Appendix G for the proof. To solve Eq. 4, for the reconstruction term, we adopt latent optimization to optimize $\theta, \{c_i\}_{i=1}^N, \{s_i\}_{i=1}^N$. For the KL term, it can not be optimised directly. However, when $\Psi$ has some specific forms, we can adopt a normalization to force each of the content embeddings $c_i$ to follow it approximately.

We can choose different forms of $\Psi$ to fit the ground truth distribution when solving the optimization problem, here we provide two examples:

**Gaussian Distribution.** Gaussian distribution is used to model the distribution of images in many works (Kingma & Welling, 2014; Kim & Mnih, 2018; Higgins et al., 2017). When setting the shared distribution $\Psi$ as a zero mean, unit variance Gaussian distribution $\mathcal{N}(0, I)$, we can use instance normalization (IN) to force each of the content embeddings $c_i$ to follow $\mathcal{N}(0, I)$ in the optimization

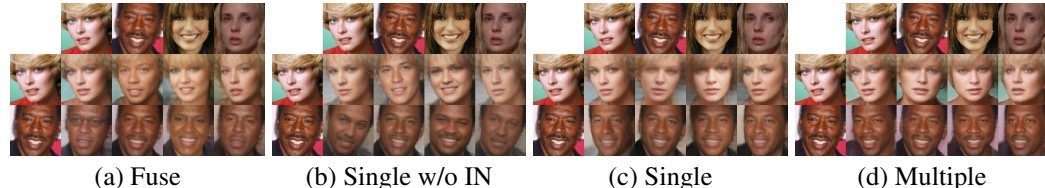

|          |          |          |          |
| :------: | :------: | :------: | :------: |
| (a) Fuse | (b) Single w/o IN | (c) Single | (d) Multiple |

Figure 2: Verification of our design for content and style. The topmost row and leftmost column provide the content and style embeddings, respectively. A good disentanglement is that: horizontally, the style (identity) of the images maintain very well when the content (pose) varies, and vertically, the content (pose) of the images align very well when the style (identity) varies.

process. By combining IN and Eq. 3, we get the same formulation as AdaIN (Huang & Belongie, 2017), which is widely adopted in style transfer tasks (Huang et al., 2018a; Kotovenko et al., 2019). Normalizing the feature map of the network to Gaussian is helpful for network training (Ioffe & Szegedy, 2015; Wu & He, 2018), but our motivation for using normalization is to force the embeddings to share the same Gaussian distribution, which differs from these works.

**Uniform Distribution.** In many datasets, the distribution of content is close to a uniform distribution, e.g., in the Chairs (Aubry et al., 2014) dataset, the images are synthesized from dense views surrounding the objects. For these datasets, we set $\Psi$ to be uniform distribution, and normalize the content embeddings with $\mathcal{L}_2$ normalization to force each of them to follow uniform distribution approximately (Muller, 1959).

As shown in Figure 1, we can use this C-S fusion block only before the generator, denoted as the Single C-S Fusion framework. We can also provide multiple paths to implement our design by inserting it before every layer of the generator, denoted as a Multiple C-S Fusion framework. For details of the network structures, please refer to Appendix A.2.

### 3.3 DEMYSTIFYING C-S DISENTANGLEMENT

In this subsection, we perform some experiments to verify that assigning different and independent roles to content and style for modeling real data distribution is the key to C-S disentanglement. The experimental setting can be found in Section 4.

If we do not assign different roles, i.e., concatenating content and style embedding as the input of the generator, the network can hardly disentangle any meaningful information for the CelebA dataset, as shown in Figure 2 (a). Our Single C-S Fusion framework can disentangle the pose and identity of human faces, as shown in Figure 2 (c). The content plays the role of modeling the shared distribution. When the shared distribution constraint is removed, i.e., without normalization, the result is shown in Figure 2 (b), where the pose and identity can not be disentangled. For the Multiple C-S Fusion framework, multiple paths are provided, and the network has more flexibility to approximate the target distribution and outperforms the Single C-S Fusion framework, as shown in Figure 2 (d).

Since the shared distribution is crucial, we experiment to demonstrate that better disentanglement can be achieved by choosing a better distribution to fit the dataset. For the real-world dataset CelebA, the distribution of pose is better modeled as a Gaussian distribution. As Figure 4 (a) and (b) show, IN achieves better disentanglement than $\mathcal{L}_2$. For the synthetic Chairs (Aubry et al., 2014) dataset, the distribution of pose is close to uniform distribution rather than Gaussian distribution. Figure 4 (c) and (d) show that the $\mathcal{L}_2$ normalization results in better identity and pose consistency.

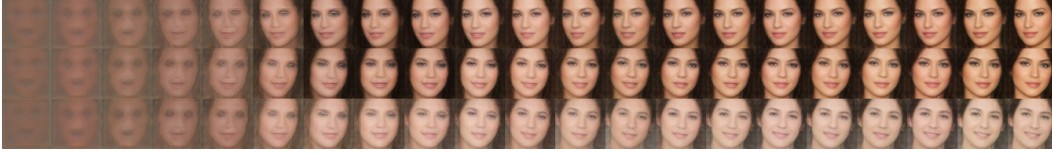

Figure 3: Generated images at different training steps. The first and second rows share the same style embedding. The second and third rows share the same content embedding.

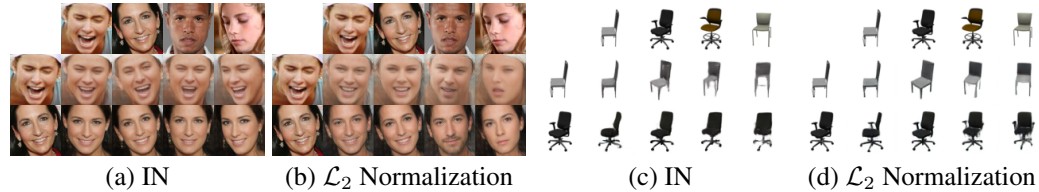

|  (a) IN | (b) $\mathcal{L}_2$ Normalization | (c) IN | (d) $\mathcal{L}_2$ Normalization |

Figure 4: Comparison of the disentanglement with different normalizations. Instance Normalization (IN) achieves better result on CelebA, e.g. the face identities are more alike with the query. $\mathcal{L}_2$ normalization outperforms on Chairs, where the shapes of chairs are more consistent in each row.

To better understand how our design helps to guide the disentanglement, we visualize the generated images during the training process in Figure 3. As the generated images show, a mean shape of faces is first learned. Then the faces start to rotate, which indicates the pose is disentangled to the content space. After that, the identity features emerge as the style starts to learn parameters for customizing the shared distribution to approximate the real faces distribution.

### 3.4 LOSS FUNCTION

**Perceptual Loss.** Perceptual Loss is widely used in weakly supervised and unsupervised methods (Wu et al., 2020; Ren et al., 2020; Wu et al., 2019c). Gabbay & Hoshen (2020) claimed that perceptual loss is not extra supervision for disentanglement. We adopt a VGG (Simonyan & Zisserman, 2015) perceptual loss $\mathcal{L}_{\mathcal{P}}$ as a reconstruction loss in Eq. 4, implemented by Hoshen et al. (2019).

**Instance Discrimination.** Instance discrimination can automatically discover appearance similarity among semantic categories (Wu et al., 2018). Inspired by this, we propose to use instance discrimination as a complementary constraint to enhance consistency among the images sharing the same style embeddings. We denote the instance discrimination loss as $\mathcal{L}_{\mathcal{ID}}$. The implementation detail can be found in Appendix C.3.

**Information Bottleneck.** Burgess et al. (2018a) propose improving the disentanglement in $\beta$-VAE by controlling the capacity increment, i.e., forcing the KL divergence to be a controllable value. This motivated us to control the information bottleneck capacity of content and style to help to avoid leakage. This loss is denoted as $\mathcal{L}_{\mathcal{IB}}$. The details of this loss are provided in Appendix C.4.

Our full objective is

$$w_P \mathcal{L}_P + w_{IB} \mathcal{L}_{IB} + w_{ID} \mathcal{L}_{ID}, \tag{5}$$

where hyperparameters $w_P$, $w_{IB}$, and $w_{ID}$ represent the weights for each loss term respectively. The ablation study for the loss terms is presented in Appendix E.

## 4 EXPERIMENTS

In this section, we perform quantitative and qualitative experiments to evaluate our method on seen data following common practice. We test our method on several datasets: **Car3D** (Reed et al., 2015), **Chairs** (Aubry et al., 2014), **CelebA** (Liu et al., 2015). For details of the datasets, please refer to Appendix B.

**Baselines.** Among all the prior works, we choose several state-of-the-art class-supervised C-S disentanglement benchmarks for comparisons: Cycle-VAE (Jha et al., 2018), a variant of VAE using cycle-consistency; DrNet (Denton & Birodkar, 2017), an adversarial approach; Lord (Gabbay & Hoshen, 2020), a latent optimization method. We also choose two unsupervised disentanglement methods: FactorVAE (Kim & Mnih, 2018), a method that encourages the distribution of representations to be factorial; Wu et al. (2019c) [1], a two-branch VAE framework based on unsupervised structure learning. More details for baselines are presented in Appendix B.

---

[1] There is no open-sourced implementation for it. We modify `https://github.com/CompVis/vunet` and provide pseudo ground truth landmarks to the network. Thus it becomes semi-supervised.

Table 1: Performance comparison in content tranfer metric ( lower is better). For Wu et al. (2019c)[1], we provide pseudo facial landmarks and have no suitable landmarks for cars and chairs.

| Method | Supervision | Cars3D | Chairs | CelebA |
|---|---|---|---|---|
| DrNet (Denton & Birodkar, 2017) | | 0.146 | 0.294 | 0.221 |
| Cycle-VAE (Jha et al., 2018) | ✓ | 0.148 | 0.240 | 0.202 |
| Lord (Gabbay & Hoshen, 2020) | | 0.089 | **0.121** | 0.163 |
| FactorVAE (Kim & Mnih, 2018) | | 0.190 | 0.287 | 0.369 |
| Wu et al. (2019c)[1] | ✗ | – | – | 0.185 |
| Ours | | **0.082** | 0.190 | **0.161** |

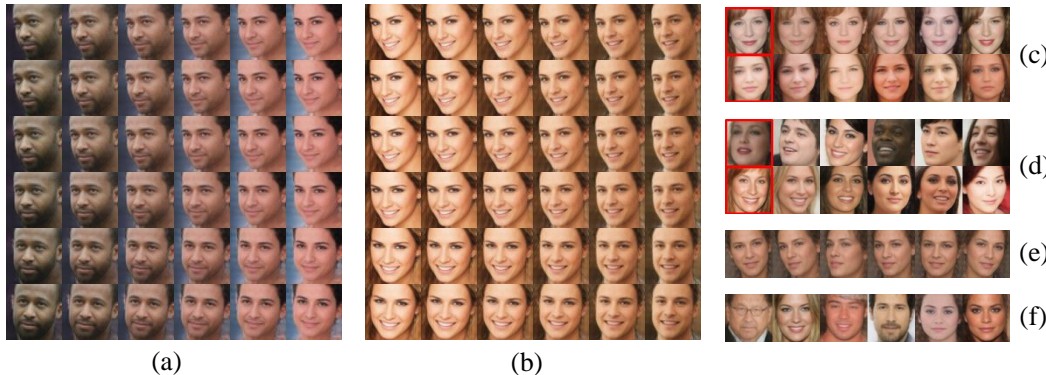

Figure 5: Demonstrations of the latent spaces by interpolation (a & b) and retrieval (c-f).

## 4.1 QUANTITATIVE EXPERIMENTS

We compare our method (Multiple C-S Fusion framework) with the baselines on Car3D, Chairs and CelebA.

**Content Transfer Metric.** To evaluate our method's disentanglement ability, we follow the protocol of Gabbay & Hoshen (2020) to measure the quality of content transfer by LPIPS (Zhang et al., 2018). Details are presented in Appendix A.1. The results are shown in Table 1. We achieve the best performance among the unsupervised methods, even though pseudo label is provided for Wu et al. (2019c). Furthermore, our method is comparable to or even better than the supervised ones.

**Classification Metric.** Classification accuracy is used to evaluate disentanglement in the literature (Denton & Birodkar, 2017; Jha et al., 2018; Gabbay & Hoshen, 2020). Following Jha et al. (2018), we train two models of a single fully-connected layer to classify content labels from style embeddings and classify style labels from content embeddings. Low classification accuracy indicates that the leakage between content and style is small. Due to no content annotations of CelebA, we regress the position of the facial landmarks from the style embeddings. The results are summarized in Table 2. Though without supervision, our method is comparable to several methods. We observe that the classification metric is also influenced by information capacity and dimensions of embeddings. For FactorVAE (Kim & Mnih, 2018), the poor reconstruction quality indicates that the latent embeddings encode a very small amount of information that can hardly be classified. The dimensions of the latent vectors of different methods vary from ten to hundreds. Actually, the higher dimension usually leads to easier classification. Based on the above observations, the classification metric may not be appropriate for disentanglement, which is also observed in Liu et al. (2020).

## 4.2 QUALITATIVE EXPERIMENTS

**Disentanglement & alignment.** In Figure 5 (a) and (b), we conduct linear interpolation to show the variation in the two latent manifolds. Horizontally, the identity is changed smoothly with the interpolated style latent space while maintaining the pose information. Vertically, the identity remains the same as the pose changes. These results illustrate the following points: 1) The learned content

Table 2: Performance comparison in classification metric (lower is better). For Wu et al. (2019c), we provide pseudo ground truth landmarks. Note that the column $(R(s) \to c)$ presents the error of face landmark regression from the style codes (higher is better).

| Method | Supervision | Cars3D | | Chairs | | CelebA | |
|---|---|---|---|---|---|---|---|
| | | $s \to c$ | $s \leftarrow c$ | $s \to c$ | $s \leftarrow c$ | $R(s) \to c$ | $s \leftarrow c$ |
| DrNet | | 0.27 | **0.03** | 0.06 | 0.01 | 4.99 | **0.00** |
| Cycle-VAE | ✓ | 0.81 | 0.77 | 0.60 | 0.01 | 2.80 | 0.12 |
| Lord | | **0.03** | 0.09 | **0.02** | 0.01 | 4.42 | 0.01 |
| FactorVAE | | 0.07 | 0.01 | 0.14 | 0.01 | 5.34 | **0.00** |
| Wu et al.[1] | ✗ | – | – | – | – | **5.42** | 0.11 |
| Ours | | 0.33 | 0.24 | 0.66 | 0.05 | 4.15 | 0.05 |

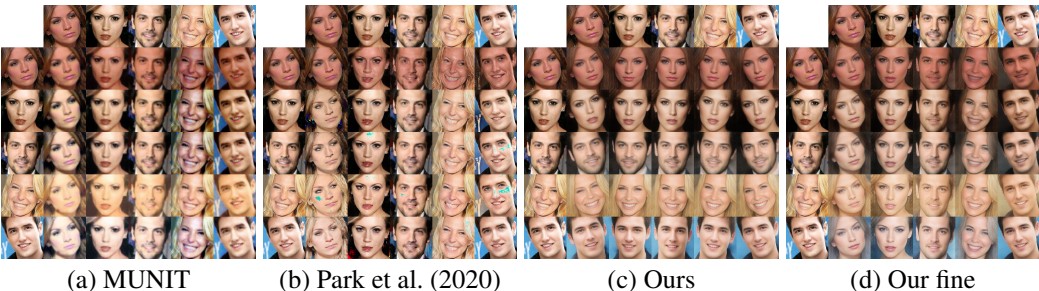

(a) MUNIT          (b) Park et al. (2020)          (c) Ours          (d) Our fine

Figure 6: Comparison with MUNIT (Huang et al., 2018b) and Park et al. (2020). MUNIT (Huang et al., 2018b) and Park et al. (2020) learn the texture information which is different from Ours (c). Our fine (d) is that we only exchange the fine styles.

and style spaces are continuous. 2) Columns of the left and right figures share the same pose, suggesting that the learned content spaces are aligned. 3) Style-related information is maintained when changing the content embedding and vice versa, suggesting the good disentanglement.

We perform retrieval on the content and style latent spaces, respectively. As shown in Figure 5 (c) and (d), the nearest neighbors in the content space share the same pose but have different identities, which reveals the alignment on content space. To better identify the faces, we let the nearest neighbors in the style space share the same pose, and the generated faces look very similar, revealing that the style is well maintained. As shown in Figure 5 (f), one interesting observation is that zero content embeddings lead to a canonical view. As we assume that the pose distribution of faces is $\mathcal{N}(0, I)$, the canonical view is the most common pose in the dataset and sampled from the peak of this distribution. We also show the faces with zero style embeddings in Figure 5 (e), and it looks like the mean face of the dataset.

**Visual Analogy & Comparison.** Visual analogy (Reed et al., 2015) is to switch between style and content embeddings for each pair within a testset. We show the visual analogy results of our method against FactorVAE (Kim & Mnih, 2018) (typical unsupervised baseline) and Lord (strongest supervised baseline) in Figure 8 on Chairs, Car3D, and CelebA. The results of FactorVAE on all datasets have poor reconstruction quality and bad content transfer. On Cars3D, Lord has artifacts (e.g., third column) and could not capture the color style of the test images (e.g., fourth row). On CelebA, the transfer result of Lord has ambiguity, e.g., the content embedding controls facial expression in the fifth column, while other content embeddings do not control expression. Our method achieves comparable pose transfer to Lord and maintains the identity of the images. For more results (including other datasets), please refer to Appendix D.

**Comparsion with Image Translation.** Starting with our assumption that content embeddings share the same distribution, and leveraging C-S fusion block, we achieve unsupervised content and style disentanglement without needing "swapping" operation and GAN loss constraint to extract the shared content information as image translation works (MUNIT (Huang et al., 2018b) and Park et al. (2020)) do. As shown in Figure 6, for MUNIT (Huang et al., 2018b) and Park et al. (2020), the

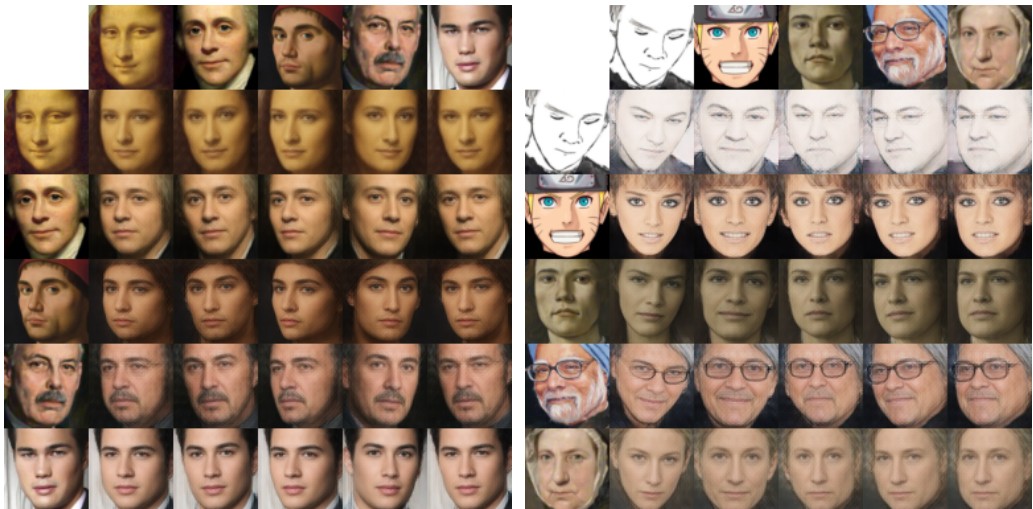

Figure 7: Inference for unseen images. Our method performs well on images from different domains: painting and cartoon.

content is low-level structure information. While in our case, the content is a high-level semantic attribute of the object, e.g., the pose attribute. As shown in Figure 6 (d), we can also achieve the similar performance to exchange the tone of the images by exchanging the fine style. The fine styles in our method are the style inputs of the last C-S fusion block in the multiple C-S fusion framework.

### 4.3 UNSEEN IMAGES INFERENCE

Though we learn to disentangle in an unsupervised manner, we may need to process unseen images. An intuitive solution is to train encoders to encode images to the latent spaces. We train style encoder $E_s$ and content encoder $E_c$ by minimizing

$$\mathcal{L}_E = \|E_s(I_i) - s_i\|_1 + \|E_c(I_i) - c_i\|_1. \tag{6}$$

We apply our model trained on the CelebA dataset to faces collected by Wu et al. (2020) including paintings and cartoon drawings. As shown in Figure 7, our method can be well generalized to unseen images from different domains.

## 5 NEW APPLICATION

In this work, we explore a new application of C-S disentanglement. For 3D reconstruction, single-view settings lack reliable 3D constraints, which can cause unresolvable ambiguities (Wu et al., 2019a). Thanks to our disentangled representations, we can generate multi-view images from a single view by extracting the style embedding of the single view and then combining it with multiple content embeddings. On Chairs, we adopt Pix2Vox (Xie et al., 2019), a framework for single-view, and multi-view 3D reconstruction to verify the advantages of our method. As shown in Figure 9, the 3D objects reconstructed from multi-view inputs generated from our method are much better than those reconstructed from a single view, and even comparable to those reconstructed from ground-truth multi-view images. For results on Celeba, please refer to Appendix D.3.

## 6 CONCLUSION

We present an unsupervised C-S disentanglement method, based on an inductive bias: assigning different and independent roles to content and style when approximating the real data distributions. Our method outperforms other unsupervised approaches and achieves comparable to or even better performance than the state-of-the-art supervised methods. We also propose to use it to help single-view 3D reconstruction, as a new application of C-S disentanglement. As for the limitation, we fail on

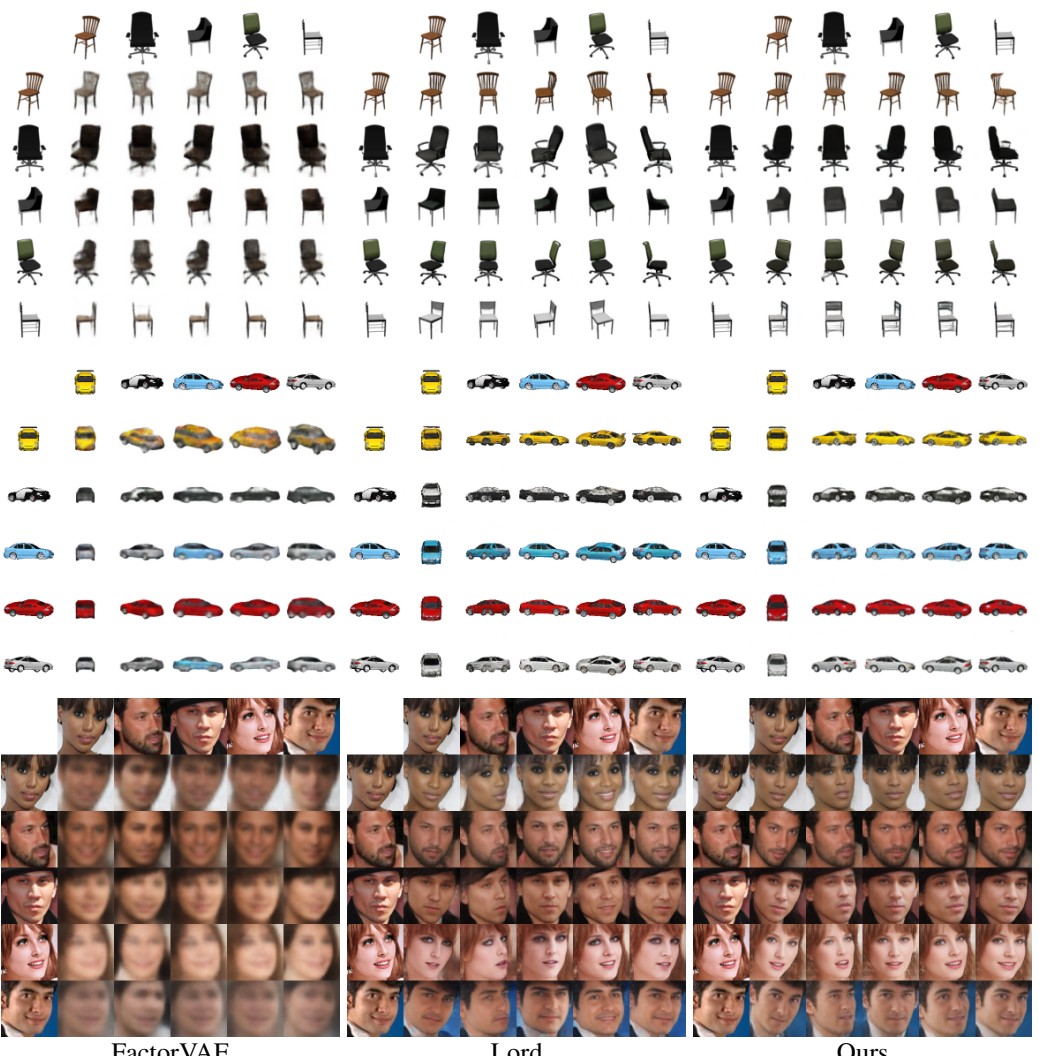

Figure 8: Comparison of visual analogy results on Chairs, Car3D and CelebA (from top to bottom).

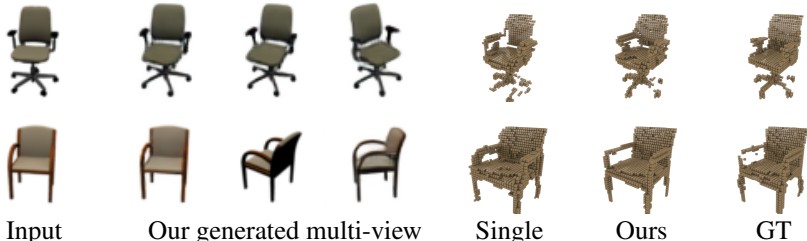

| Input | Our generated multi-view | Single | Ours | GT |

Figure 9: 3D reconstruction results on Chairs. We generate multi-view from Input. Single: the object reconstructed by only Input. Ours: the object reconstructed by multi-view inputs. GT: the object reconstructed by the ground truth of multi-view inputs.

datasets containing multiple categories with large appearance variation, e.g., CIFAR-10 (Krizhevsky et al., 2009), which does not match our assumption. Our method could be adopted to help downstream tasks, e.g., domain translation, ReID, etc. An interesting direction is to apply our method to instance discrimination. With disentangled representations, contrastive learning is expected to perform more effectively.

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

## A    IMPLEMENTATION DETAILS

All the models are trained on an Nvidia Tesla V100 GPU. The model is implemented in Py-Torch (Paszke et al., 2017). For Car3D, Chairs, and Celeba, we set the size of the style embedding $d_s$ to 256, the size of the content embedding $d_c$ to 128, and the epoch to 200. All the images are resized to $64 \times 64$. For the hyperparameters, $w_P = 5, w_{IB} = 1, w_{ID} = 1$. We used Adam (Kingma & Ba, 2015) with a learning rate of 0.0003 for the models and 0.003 for the latent spaces.

### A.1    EXPERIMENTAL DETAILS

**Content Transfer Metric.** For Car3D and Chairs datasets, the content labels (ground truth) are available. For images $I_i$ and $I_j$ sampled from the testing set randomly, we compute LPIPS between $G_\theta(c_i, s_j)$ and the corresponding ground truth from the same class of $I_j$. For CelebA, we randomly sample two images $I_i$ and $I_j$ with the same identity, and we retrieve an image that has the most similar pose $k$ from the test set, i.e., the nearest neighbor in the 68 facial-landmarks space. We measure the similarity between $I_j$ and $G_\theta(c_i, s_k)$.

## A.2 NETWORK STRUCTURE

Our Single C-S Fusion framework and Multiple C-S Fusion framework are shown in Figure 10. For the details of the reparametric module $R$, please refer to Appendix C.2.

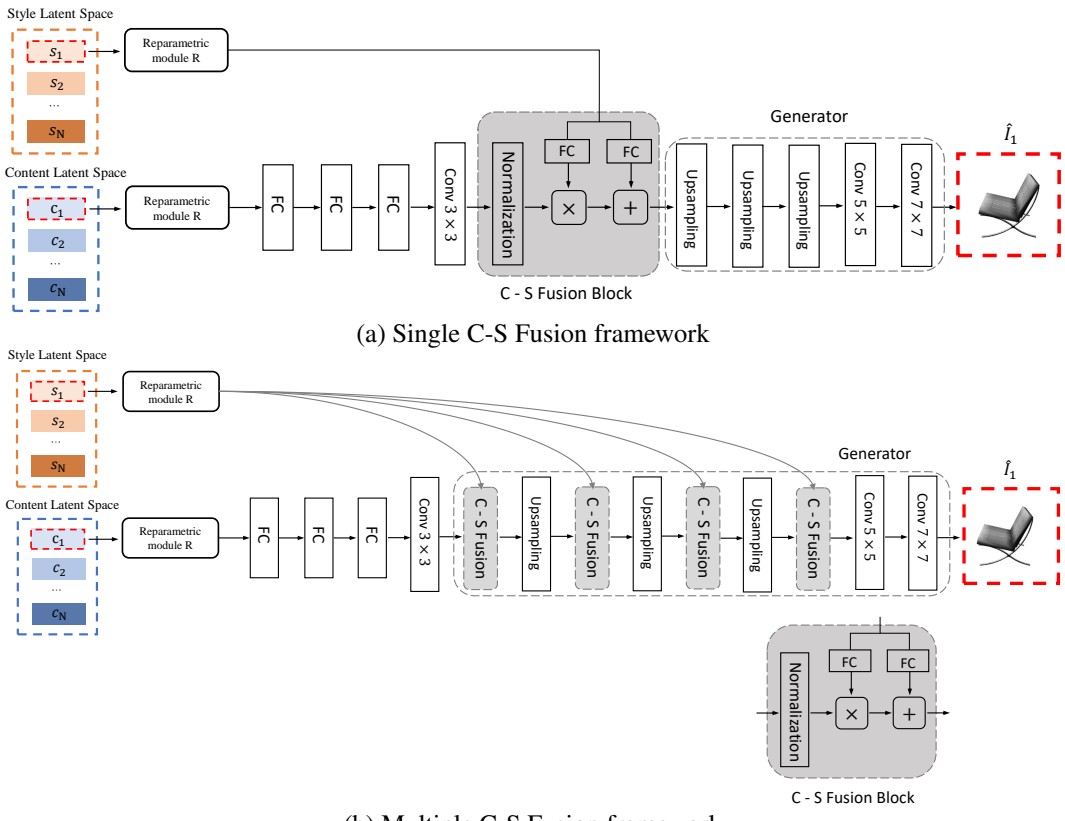

(a) Single C-S Fusion framework

(b) Multiple C-S Fusion framework

Figure 10: Details of network structure. For every upsampling layer, there is a $3 \times 3$ convolutional layer following it. The embedding $s_1$ and $c_1$ from style and content embedding space respectively are first processed by the reparametric model $R$, then fed into the network in different ways. Then the image $\hat{I}_1$ is generated. The style and content latent space and network are jointly optimized under the supervision of the reconstruction loss between synthesized image $\hat{I}_1$ and ground truth image I1 from the dataset.

## B BASELINE DETAILS

For the datasets in the main paper, Car3D contains 183 car models, each rendered from 96 poses. Chairs consists of 1393 chair models, each rendered from 62 poses. CelebA contains 202,599 facial images of 10,177 celebrities.

For the baselines, we use open-sourced implementations for Cycle-VAE (Jha et al., 2018) [2], Dr-Net (Denton & Birodkar, 2017) [3], Lord (Gabbay & Hoshen, 2020) [4] and FactorVAE (Kim & Mnih, 2018) [5].

For FactorVAE, we traverse the latent space to select the dimensions related to pose as content embedding and treat the other dimensions as style embedding. For Wu et al. (2019c), there is no open-sourced implementation. We use the code from `https://github.com/CompVis/vunet`,

---

[2] `https://github.com/ananyahjha93/cycle-consistent-vae`
[3] `https://github.com/ap229997/DRNET`
[4] `https://github.com/avivga/lord-pytorch`
[5] `https://github.com/1Konny/FactorVAE`

which uses ground truth landmarks as input instead of learning the landmarks unsupervisedly. To achieve the pseudo ground truth landmarks, we use the face detection library (Bulat & Tzimiropoulos, 2017) for Celeba. We try to use the L1 and perceptual loss for all the baselines and select the best.

We split the datasets into training and testing sets. For Celeba, we randomly select 1000 among 10177 celebrities for testing. For Car3D, we randomly select 20 among 183 CAD models for testing. For Chairs, we randomly select 100 among 1393 models for testing. For baselines with group supervision, only the training sets are used for training. For unsupervised baselines and our method, all the datasets are used for training.

## C  TECHNICAL COMPONENTS

Here we present three technical components that are helpful to the C-S disentanglement. The ablation study for these components is shown in Appendix E.

### C.1  LATENT OPTIMIZATION.

In the C-S disentanglement literature, it is common to use encoders to predict embeddings, while latent optimization (Bojanowski et al., 2018; Gabbay & Hoshen, 2020) directly optimizes the embeddings via back-propagating without using encoders. Encoders have a large number of parameters and require a lot more extra effort for training. Therefore, We adopt the latent optimization approach to update the latent spaces directly.

### C.2  REPARAMETRIC MODULE

Inspired by VAE (Kingma & Welling, 2014), we design a reparametric module to force the latent space to be continuous. Thus, the embeddings encoding similar information will get closer in the latent space. Assume we have a mean embedding $\mu$ with a standard deviation $\sigma$, the reparametrized output is $\sigma X + \mu$, where $X \sim \mathcal{N}(0, I)$. To further simplify the problem, we set $\sigma = 1$ following Wu et al. (2019c) and Gabbay & Hoshen (2020). The mean embedding is the input style or content embedding. The reparametric module can make the latent space continuous, which is helpful for backpropagation. Though the training images have discrete identities, the optimized style embedding space is continuous. Different style embeddings of the people with similar appearances are close to each other, as shown in Figure 5 (c).

### C.3  INSTANCE DISCRIMINATION LOSS

We first pretrain a ResNet-18 (He et al., 2016) $\Phi$ and define a collection of layers of $\Phi$ as $\{\Phi_l\}$. Among several representative methods (Wu et al., 2018; Ye et al., 2019; He et al., 2020), we observe that the method in Wu et al. (2018) achieves the best performance in our task. Given two images $I_i$ and $I_j$, we mix the embeddings to generate $u = G(R(s_i), R(c_j))$ and $v = G(R(s_j), R(c_i))$. For samples sharing the same style embedding, we enforce the feature distance in $\Phi$ between them to be close. This loss term can be written as

$$\mathcal{L}_{\mathcal{ID}} = \sum_l \lambda_l (\|\Phi_l(u) - \Phi_l(x)\|_1 + \|\Phi_l(v) - \Phi_l(y)\|_1), \tag{7}$$

where $x = G(R(s_i), R(c_i))$ and $y = G(R(s_j), R(c_j))$. The hyperparameters $\{\lambda_l\}$ balance the contribution of each layer $l$ to the loss. $\{\lambda_l\}$ are set to be $[1, 1, 1, 1, 1]$

### C.4  INFORMATION BOTTLENECK

Similar to Anneal VAE (Burgess et al., 2018a), we introduce a information bottleneck given by

$$\mathcal{L}_{\mathcal{IB}} = \gamma_s \|s^2 - C_s\|_1 + \gamma_c \|c^2 - C_c\|_1 \tag{8}$$

where $C_s$ and $C_c$ are the information capacity controlling the amount of information of the content and style respectively. During training, $C_s$ and $C_c$ increase linearly. The rate of increase is controlled by the increase steps and the maximum value. By controlling the increase rate, the content is forced

to encode information first, so that the learning process is more consistent with our assumptions about the data: the shared conditional variable $c$ is learned first.

For the information bottleneck, by taking the training process of the model without the information bottleneck as a reference, we determine the increase steps and the maximum of the information capacity $C_c$ and $C_s$. We can enhance the model inductive bias by tuning these parameters. For Chairs, we set the maximum of $C_c$ to 5, the start value of $C_c$ to 2, the increase steps of $C_c$ to $1.4 \times 10^5$, $\gamma_c$ to 1 and $\gamma_s$ to 0. Note that our model achieves state-of-the-art performance on Chairs even without information bottleneck.

## D  MORE RESULTS

In this section, we demonstrate more qualitative comparison and more qualitative results (including more datasets).

### D.1  MORE QUALITATIVE EXPERIMENTS

In the main paper, for unsupervised baselines, we only compare our method with FactorVAE (Kim & Mnih, 2018) limited to space. As shown in Figure 11, we also outperform Wu et al. (2019c). For Wu et al. (2019c), the disentanglement is poor, such that the content embeddings control almost all the factors while the style embeddings control the tone.

For datsets in the main paper, We provide more qualitative results in Fig. 24, 25, 26, 27 and 28. Moreover, we also apply our method on higher resolution images and achieve good performance, as shown in Figure 20.

### D.2  MORE DATASETS

Besides the datasets introduced in the main paper, we make additional experiments on other datasets: such as **MNIST** (LeCun et al., 2010), **Cat** (Parkhi et al., 2012; Zhang et al., 2008), **Anime** (Chao, 2019) and Market-1501 (Zheng et al., 2015). MNIST has 70k examples for 10 handwritten digits. Cat has 1.2k cat head images. Anime contains 63,632 anime faces. Market-1501 have 25,259 images. The results are shown in Figure 21, 22 ,23. Furthermore, we show our results on the Market-1501 dataset in Figure 19, which demonstrates our method can disentangle the human pose and the appearance even though the skeletons have large variances.

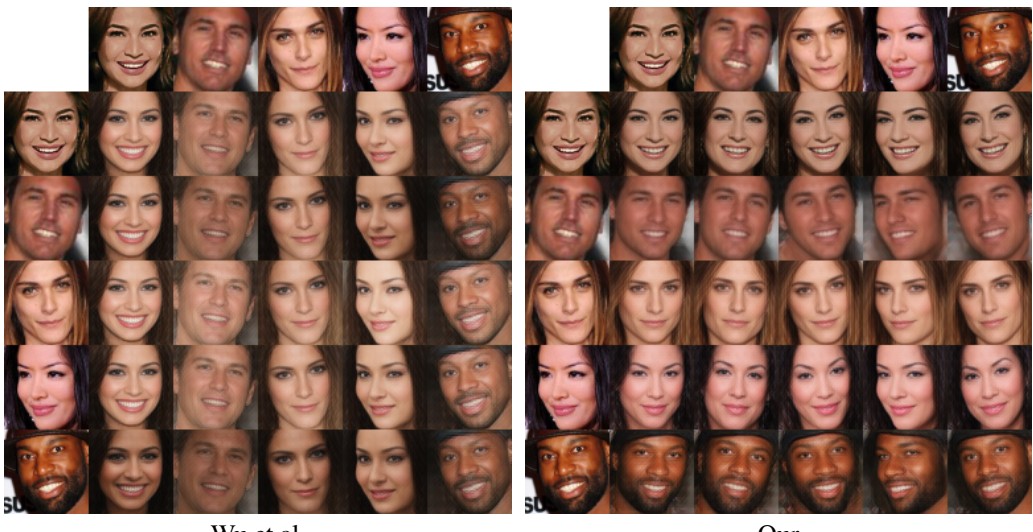

Wu et al.            Our

Figure 11: Comparison between Wu et al. and our method. For Wu et al., the images are mainly determined by content embeddings, while style embeddings only change the tone.

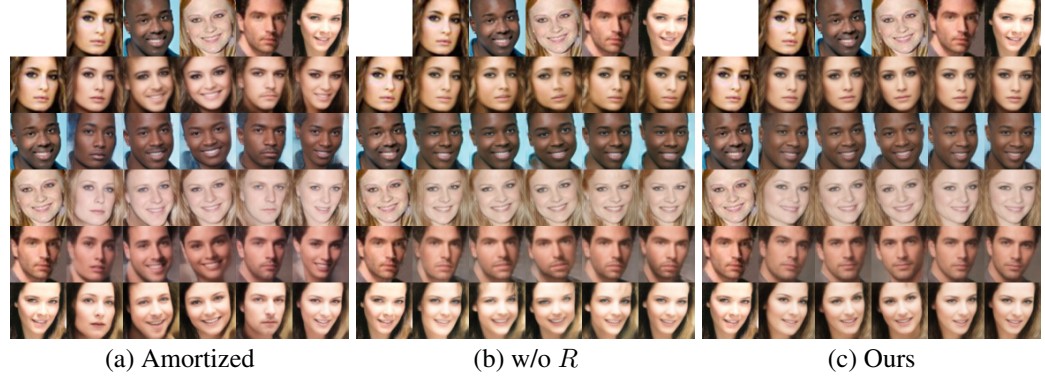

(a) Amortized        (b) w/o $R$        (c) Ours

Figure 12: Ablation study. $R$ indicates reparametric module.

### D.3 MORE 3D RECONSTRUCTION

Our setting treats every image as a single identity (style) without ambiguity for augmenting single-view images. On Celeba, We use MVF-Net (Wu et al., 2019a) based on multi-view to reconstruct 3D facial shapes. For a given image, we can get the corresponding style embedding content embedding. Then we can get the front, left, and right view of this image combining the extracted style embedding and prepared content embeddings [6]. As shown in Figure 13, our augmented multi-view images are consistent, and the 3D meshes based on our method are more accurate than those based on Lord.

## E MORE ABLATION STUDY

Here we perform more ablation study for the technical modules.

If we use an amortized scheme instead of a latent optimization scheme, there are leaks between style and content latent space, and the result is worse than latent optimization, as shown in Figure 12 (a) and (c). Furthermore, if we do not use a reparametric module, we find the reconstruction performance is worse, as shown in Figure 12 (b). For the instance discrimination loss, the comparison is shown in Table 4. The disentanglement is better with an instance discrimination loss. For the information bottleneck, as shown in Table 3, the result with an information bottleneck is much better than the one without it.

Table 3: Ablation study for infomation bottleneck on Chairs dataset. Lower is better.

| Method | Content transfer |
| --- | --- |
| Ours (w/o Information Bottleneck) | 0.280 |
| Ours | 0.190 |

Table 4: Ablation study for instance discrimination on Celeba dataset. Lower is better.

| Method | Content transfer |
| --- | --- |
| Ours (w/o Instance Discrimination) | 0.165 |
| Ours | 0.161 |

---

[6]We retrieve the nearest neighborhood (facial landmarks space) of suggested inputs of MVF-Net and extract content embedding.

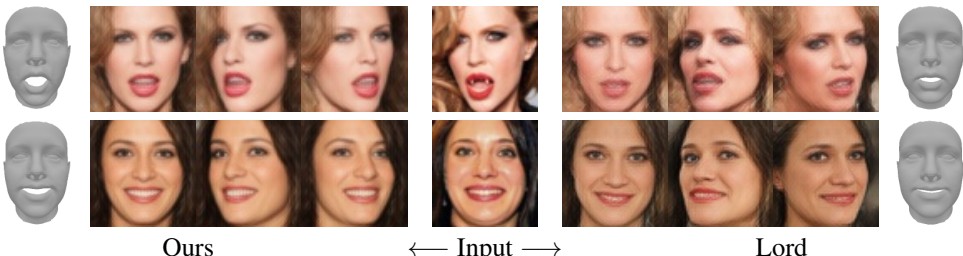

Ours      ⟵ Input ⟶      Lord

Figure 13: 3D face reconstruction. Given an image, we first generate multi-view images and then use them as augmented input.

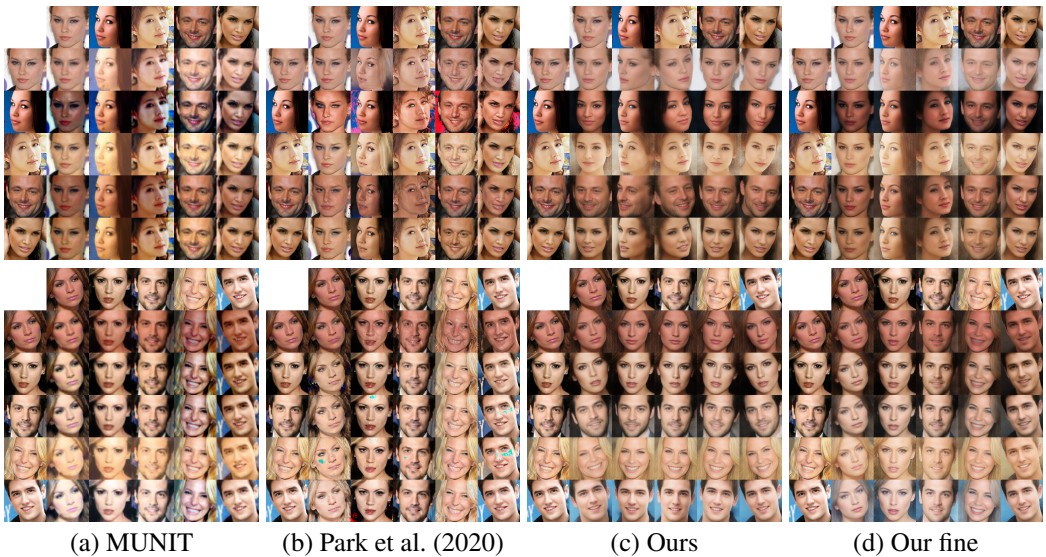

(a) MUNIT      (b) Park et al. (2020)      (c) Ours      (d) Our fine

Figure 14: Comparison with MUNIT (Huang et al., 2018b) and Park et al. (2020). MUNIT (Huang et al., 2018b) and Park et al. (2020) learn the texture information which is different from Ours (c). Our fine (d) is that we only exchange the fine styles.

## F  COMPARISON WITH SELECTED RELATED WORK

**Comparison with StyleGAN.** In our framework, the optimized content (conv) and style embeddings are disentangled representations of corresponding images. While StyleGAN (Karras et al., 2019) keeps the input of the convolution branch as a learnt constant for the whole dataset and finds the feature space of the "style" branch has disentanglement ability. For StyleGAN2 (Karras et al., 2020) [7], we select the subset of "style", which represents pose, as the content embedding and the rest subset as the style embedding. As shown in Figure 15, StyleGAN2 entangled pose with other semantic attributes, such as hair and glasses. As shown in Figure 28, the content of our method on human faces is pose attribute without entanglement.

**Comparsion with MUNIT & Park et al. (2020).** Starting with our assumption that content embeddings share the same distribution, and leveraging AdaIN-like operation, we achieve unsupervised content and style disentanglement without needing "swapping" operation and GAN loss constraint to extract the shared content information as image translation works (MUNIT (Huang et al., 2018b) and Park et al. (2020)) do. As shown in Figure 14, for MUNIT (Huang et al., 2018b) and Park et al. (2020), the content is low-level structure information. While in our case, the content is a high-level semantic attribute of the object, e.g., the pose attribute. As shown in Figure 14 (d), we can also achieve the similar performance to exchange the tone of the images by exchanging the fine style.

---

[7]We use the implementation from `https://github.com/rosinality/stylegan2-pytorch`.

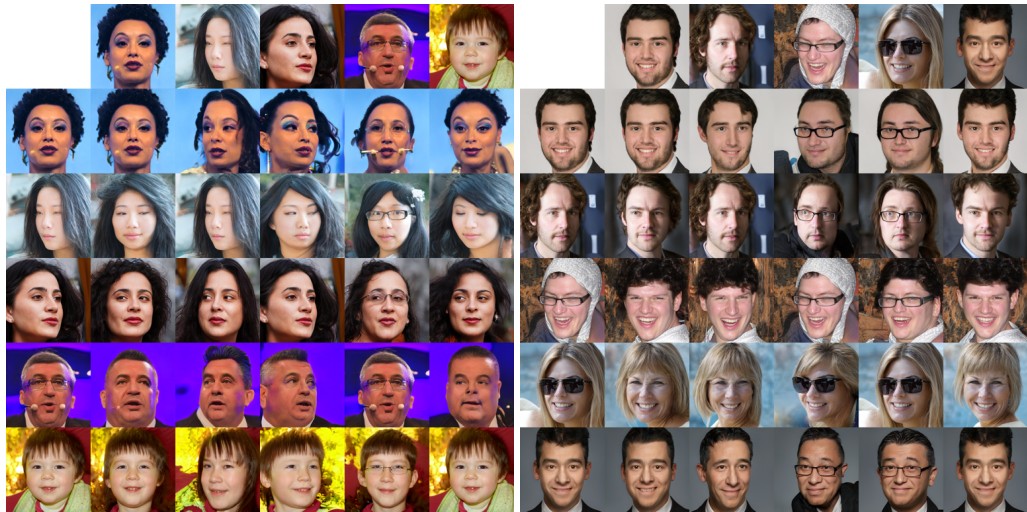

StyleGAN2

Figure 15: Performance of StyleGAN2 (Karras et al., 2020) on human faces. For StyleGAN2, the content contains entangled semantic attribute, such as pose, hair and glasses. In our case, the content is pose, which is a high-level semantic attribute of the object.

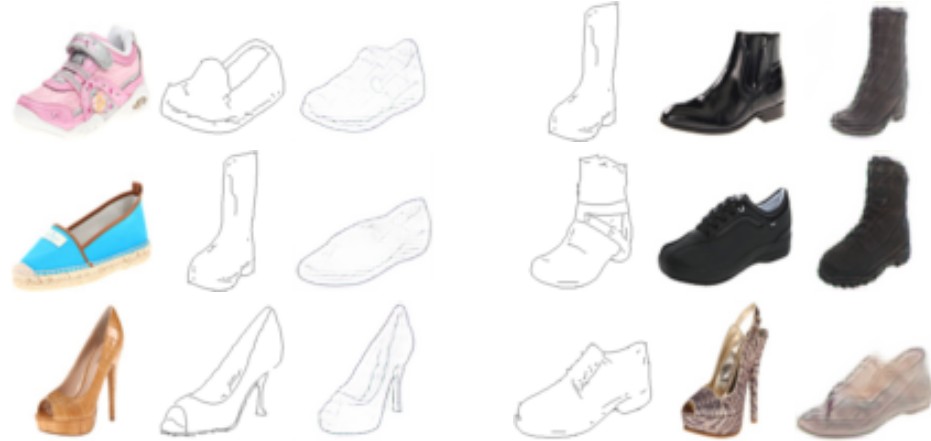

Figure 16: Examples of translating shoes to edge (left column) and translating edges to shoes (right column). Triplet order (left to right) is: content, style, translation.

The fine styles in our method are the style inputs of the last C-S fusion block in the multiple C-S fusion framework.

## G    CROSS-DOMAIN APPLICATION

As shown in the main paper, the content and style are disentangled in a single domain. Based on our assumption, the cross-domain dataset also can be disentangled. In this section, we test our model on a cross-domain dataset to further verify our assumption. In some cases that we merge images from two domains, our method can still work and achieve performance, which is similar to domain translation. For example, **Edges2Shoes** (Yu & Grauman, 2014) is a dataset consisting of 50k paired shoe and edge map images. As shown in Figure 16, the content is edge structure, and the style is texture. Thanks to this, we can translate edge images into shoe images and vice versa without any additional operation.

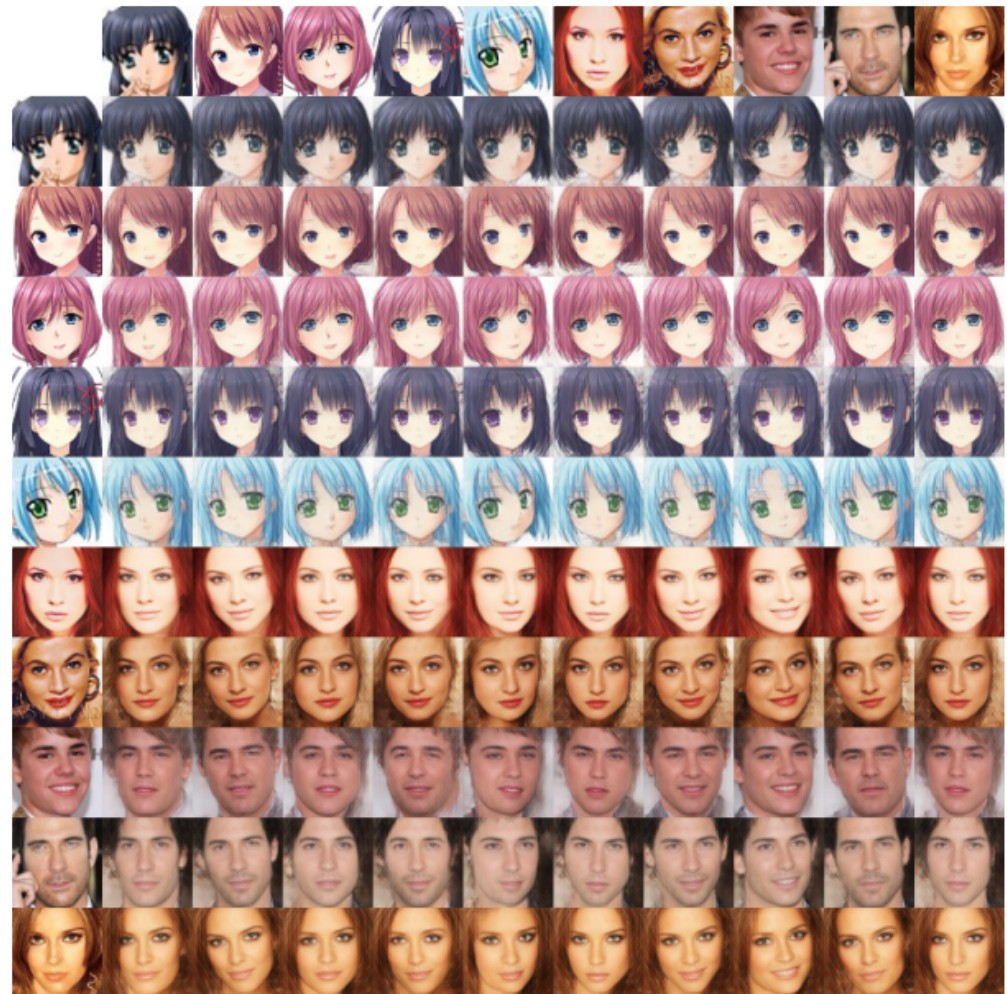

Figure 17: Results of modified model on merged cross-domain dataset based on Celeba and Anime. The learned content embedding are well aligned both in the animation and reality domain.

Furthermore, once the domain labels are given, we can disentangle and align the cross-domain dataset. This experiment may be helpful for domain transfer and domain adaptation. We train our model on the dataset that consists of Celeba and Anime. The model needs to be modified for learning cross-domain data: concatenate the domain embedding and the style embedding, take it as the style embedding in the original model, and optimize the domain embedding during latent optimization. The results are shown in Figure 17. The learned poses are well aligned both in the animation and reality domain.

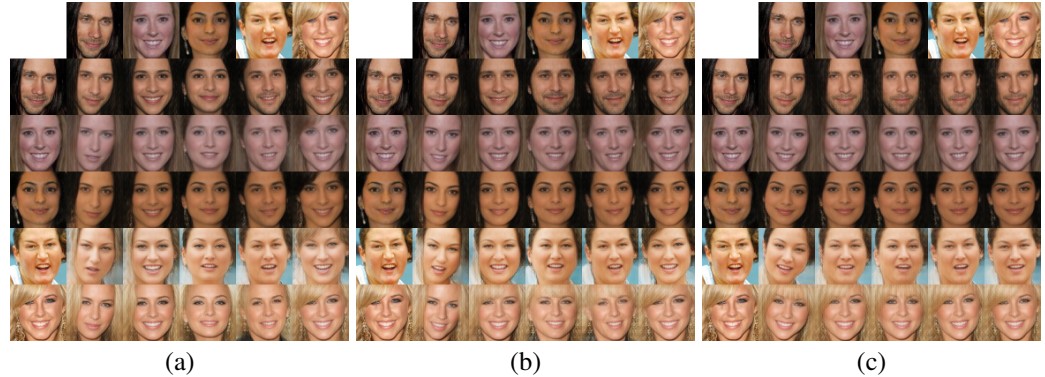

(a)  (b)  (c)

Figure 18: Study on the influence of size of embeddings. For (a), we set the size of the style embedding $d_s$ to be 128, the size of the content embedding $d_c$ to be 256. The content embeddings contain shape of face, facial expression and pose. For (b), $d_s = 256$ and $d_c = 256$, and the content embeddings contain shape of face and facial expression. For (c), $d_s = 256$ and $d_c = 128$, which is the setting used in our paper, the content embeddings contain pose.

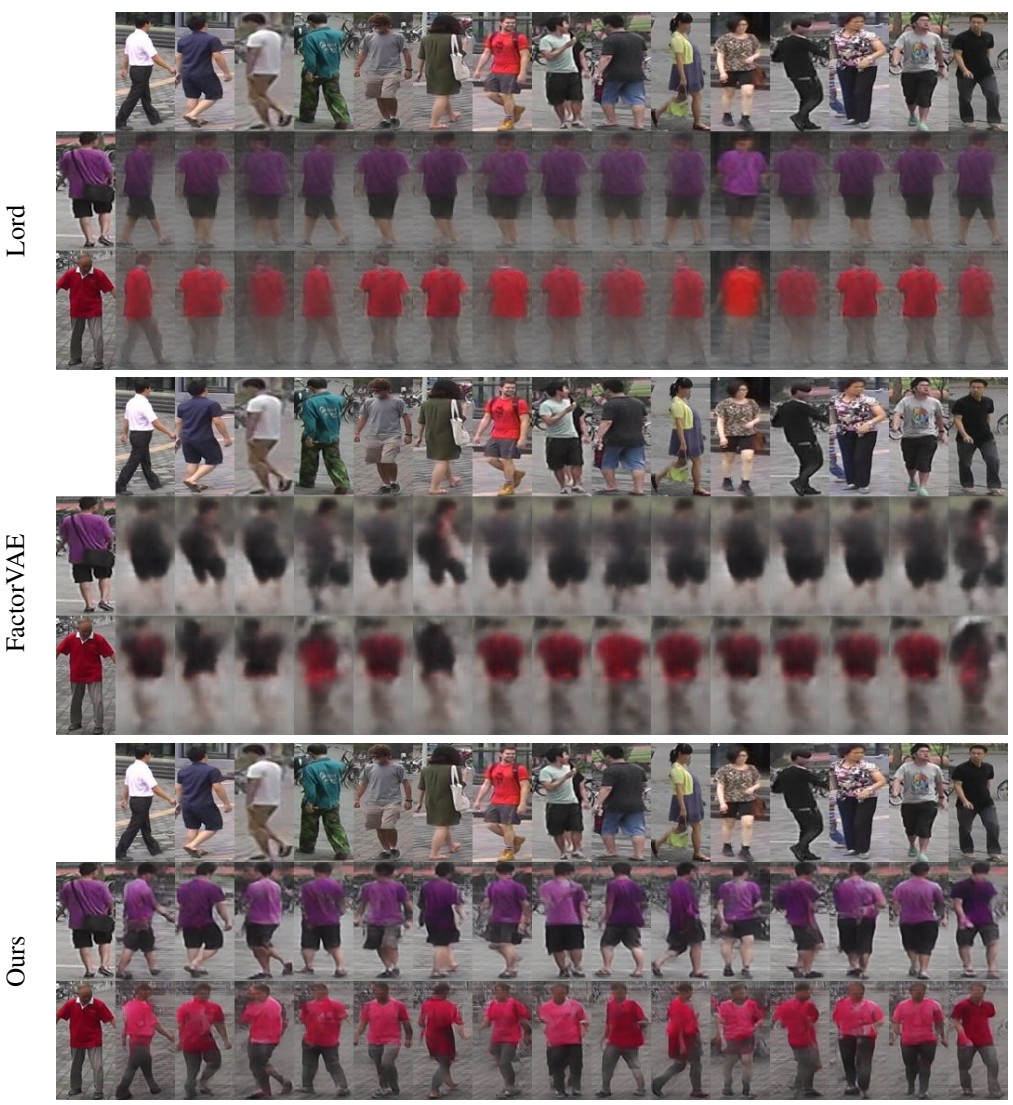

Figure 19: Comparison of visual analogy results on Market-1501 dataset. Our method outperforms supervised method Lord Gabbay & Hoshen (2020) and unsupervised method FactorVAE Kim & Mnih (2018) significantly.

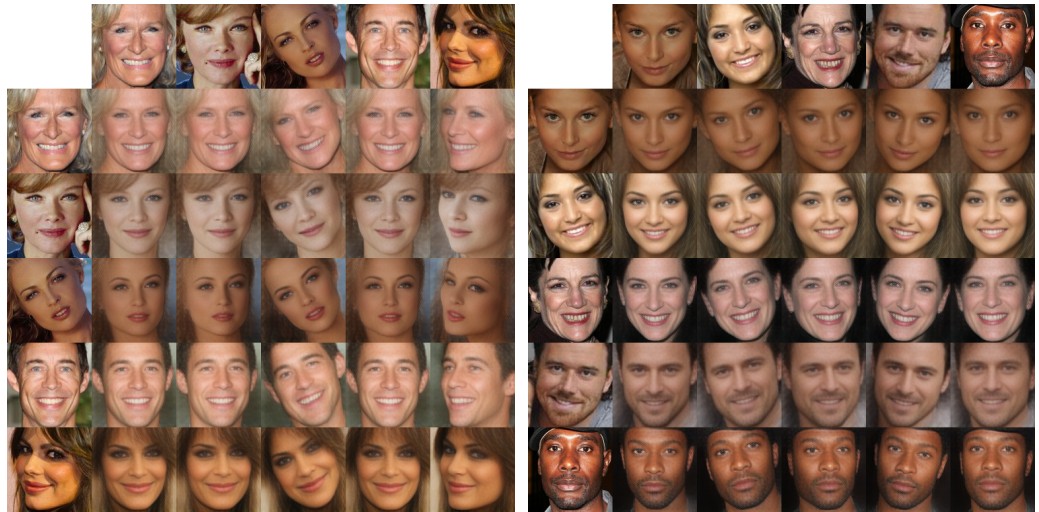

Figure 20: Results on Celeba with $128 \times 128$ resolution. Zoom in for better view.

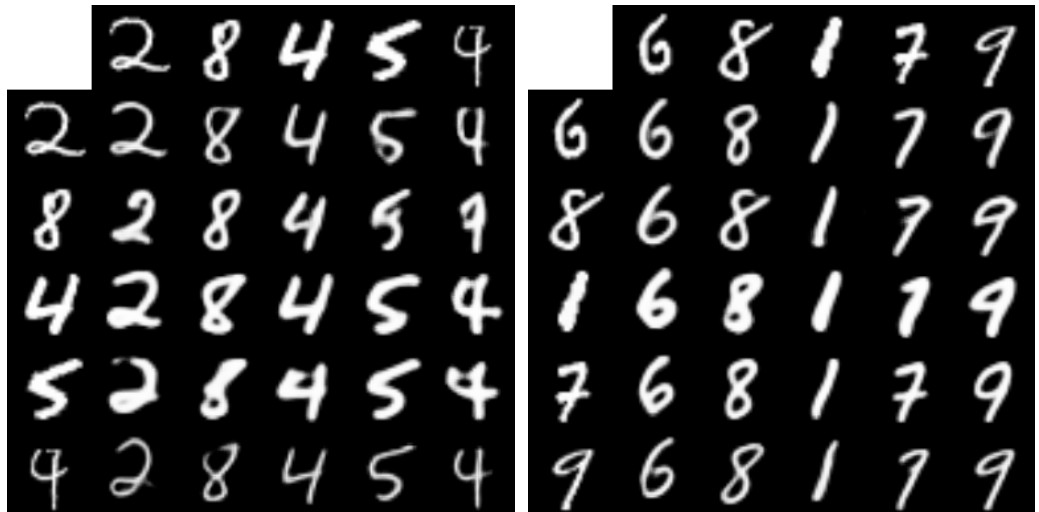

Figure 21: Results on the MNIST dataset. Content indicates geometric attributes, and style indicates texture.

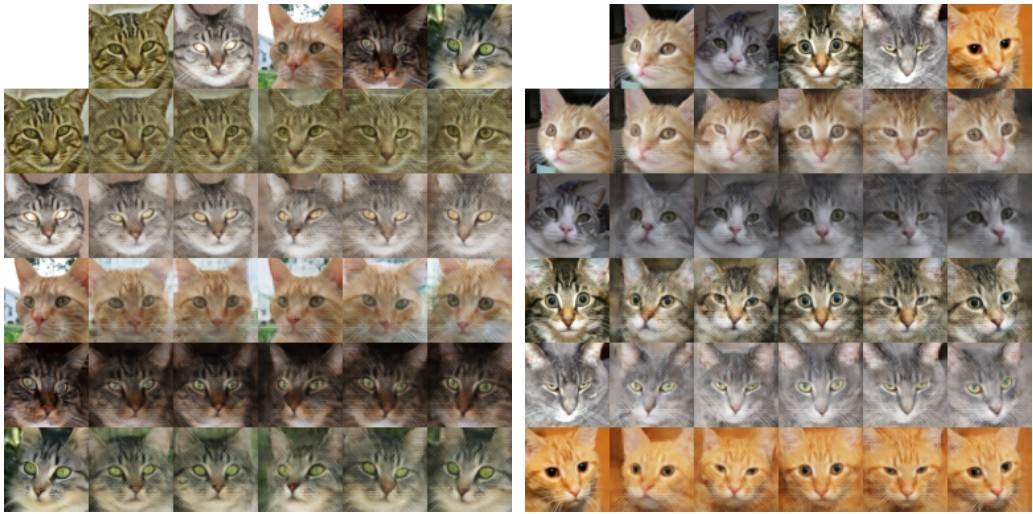

Figure 22: Results on the Cat dataset. Content indicates pose, and style indicates identity. The result further qualitatively demonstrates the ability of our method to disentangle on real-world data.

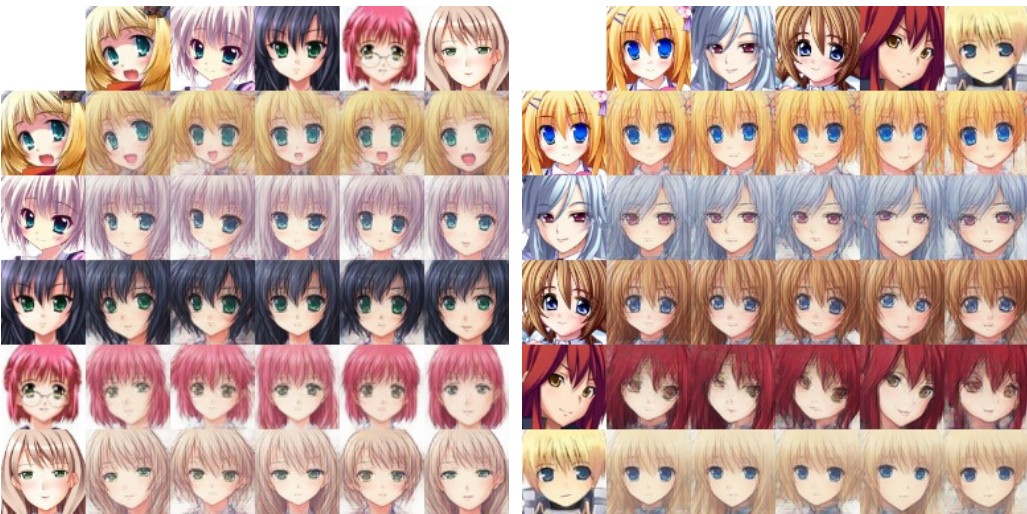

Figure 23: Results on the Anime dataset. Content indicates pose, and style indicates identity.

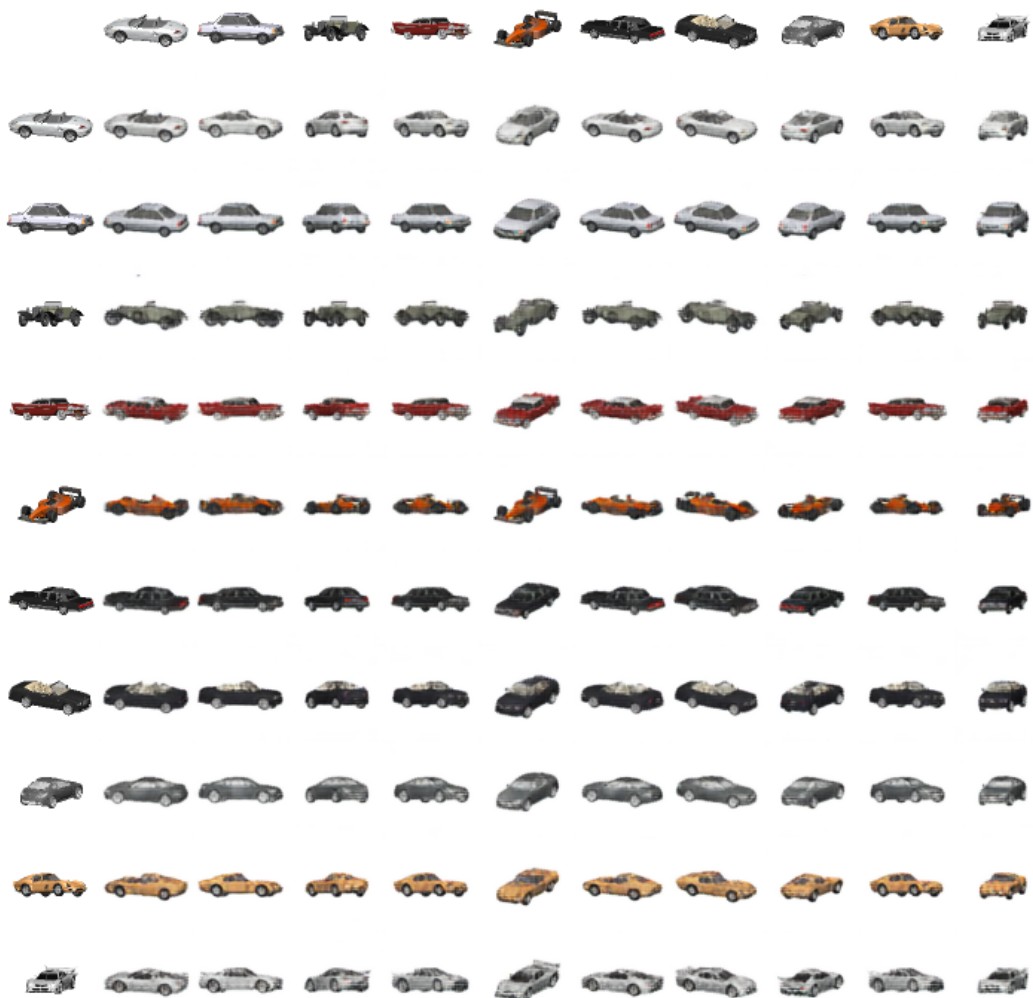

Figure 24: More visual anology of our method on Car3D.

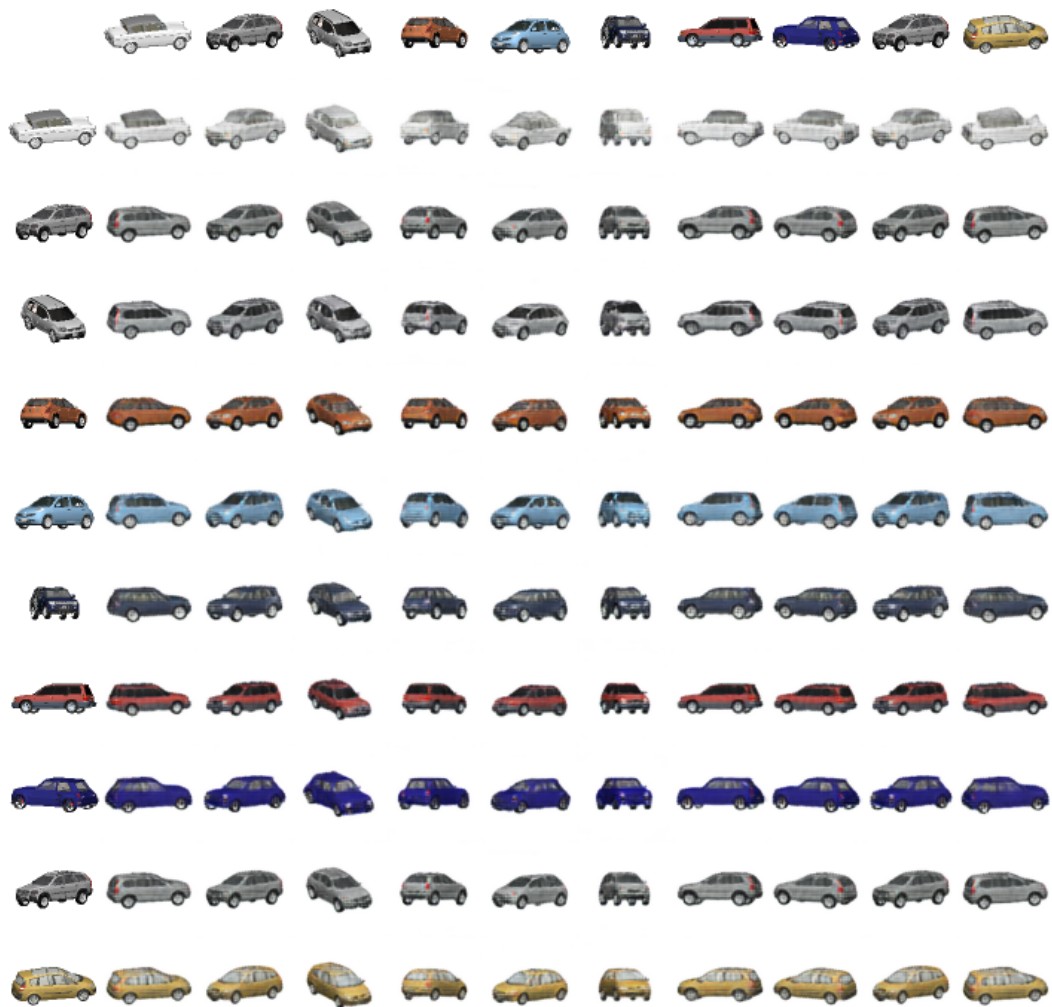

Figure 25: More visual anology of our method on Car3D.

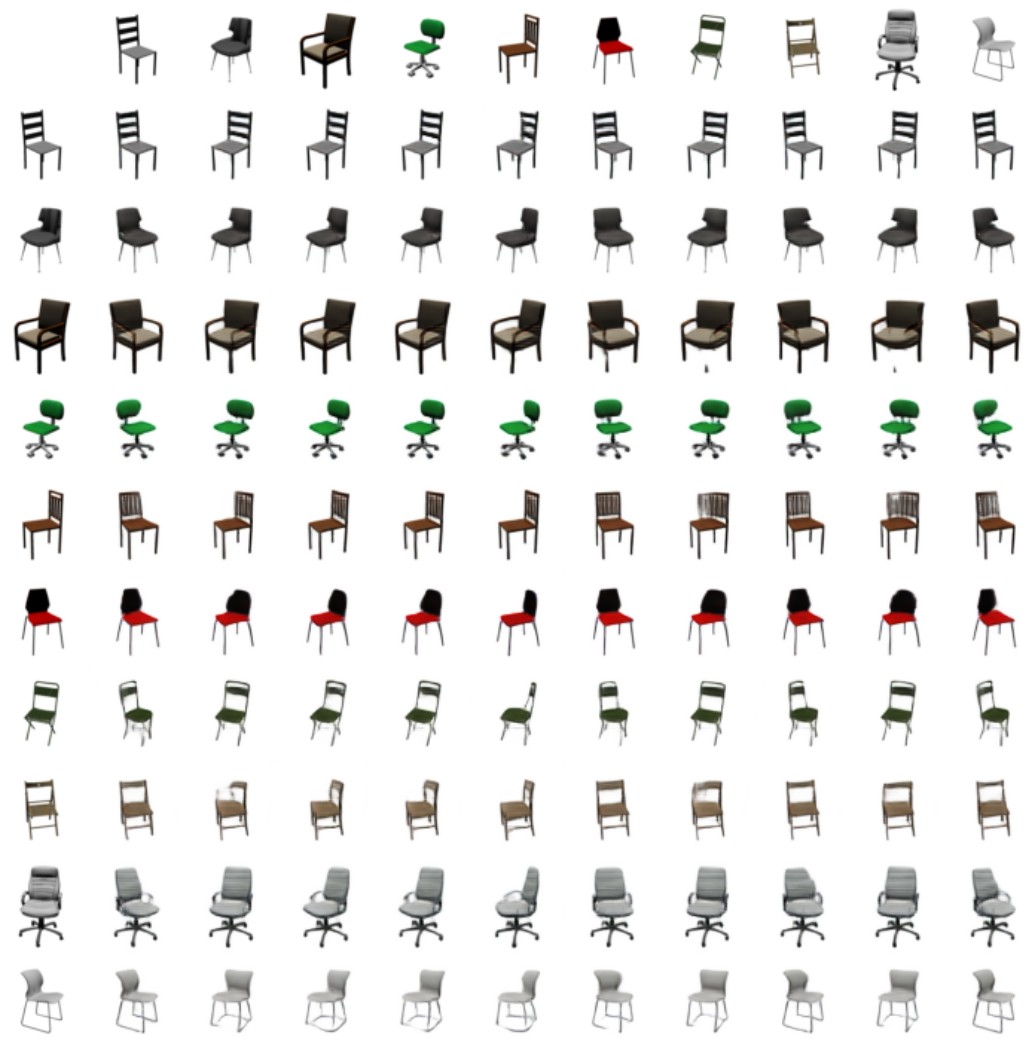

Figure 26: More visual anology of our method on Chairs.

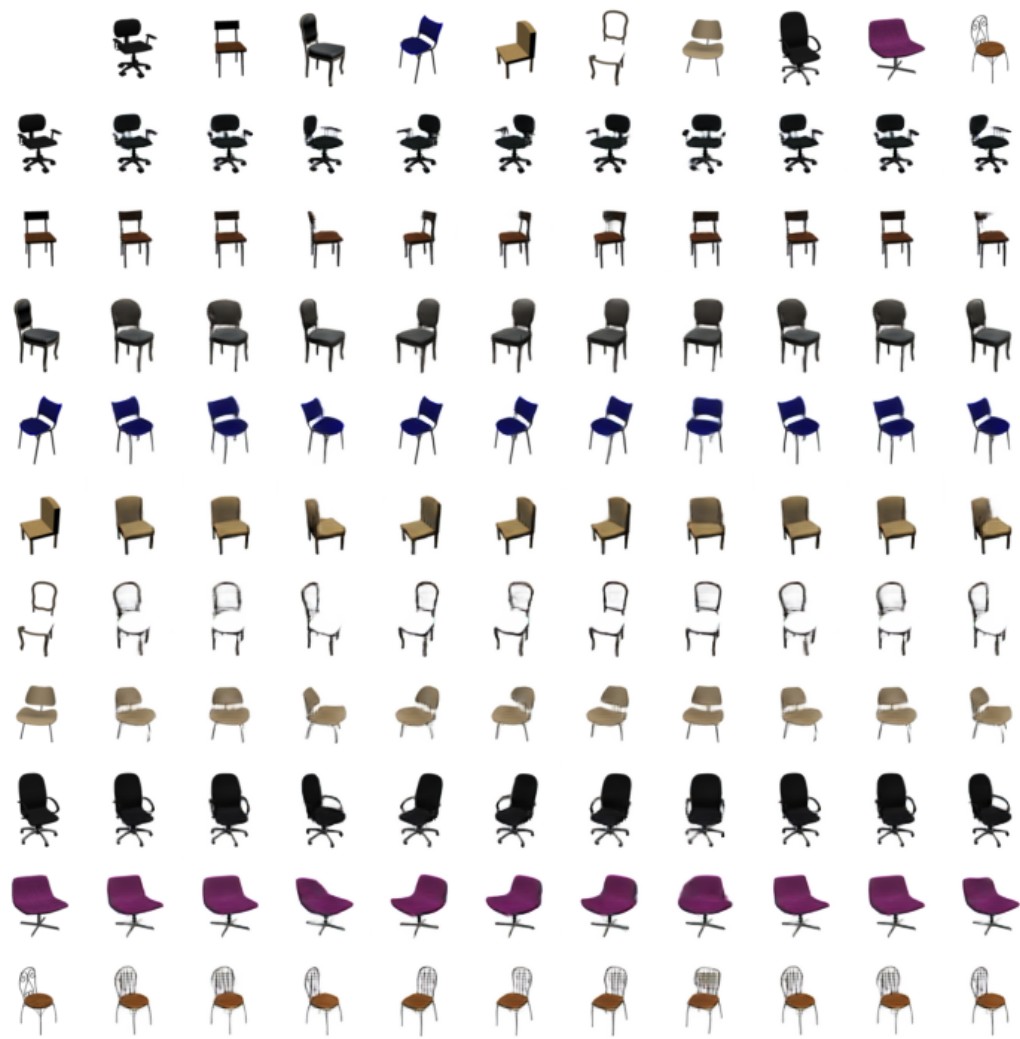

Figure 27: More visual anology of our method on Chairs.

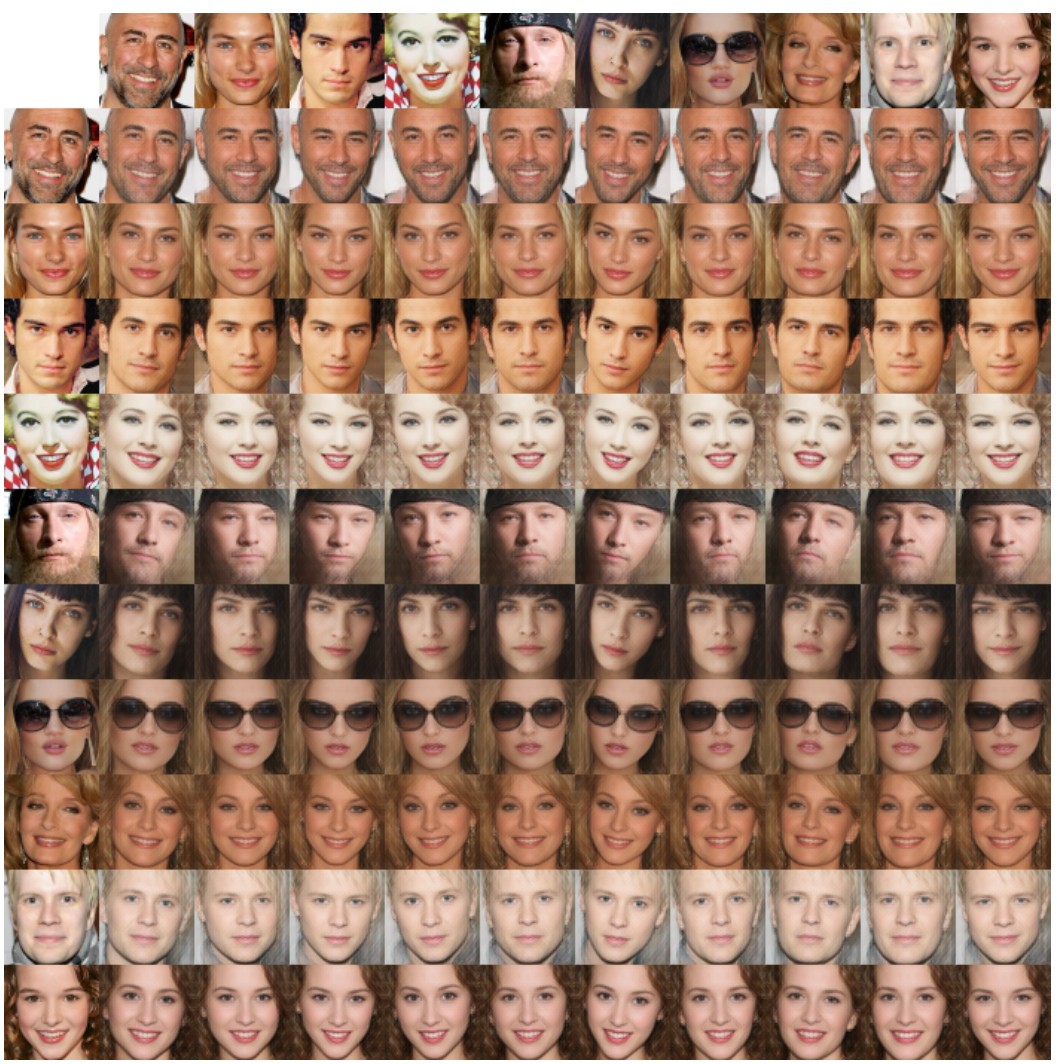

Figure 28: More visual anology of our method on Celeba.

# H PROOF

Our optimization target is to minimize the KL divergence between P and Q,

$$\min_{\theta, c_i, s_i} \sum_{i=1}^{N} KL(P_i(\boldsymbol{x}|\boldsymbol{c} = c_i) || Q_{\theta, s_i}(\boldsymbol{x}|\boldsymbol{c} = c_i)). \tag{9}$$

Expanding the above KL term, we have,

$$\min_{\theta, c_i, s_i} \sum_{i=1}^{N} \int_x P_i(\boldsymbol{x}|\boldsymbol{c} = c_i) \log \frac{P_i(\boldsymbol{x}|\boldsymbol{c} = c_i)}{Q_{\theta, s_i}(\boldsymbol{x}|\boldsymbol{c} = c_i)} dx. \tag{10}$$

The above integral equation cannot be directly calculated, but can be estimated by the sampled images $\{I_j\} \sim P_i$,

$$\min_{\theta, c_i, s_i} \sum_{i=1}^{N} \sum_{I_j \sim P_i} P_i(\boldsymbol{x} = I_j|\boldsymbol{c} = c_i) \log \frac{P_i(\boldsymbol{x} = I_j|\boldsymbol{c} = c_i)}{Q_{\theta, s_i}(\boldsymbol{x} = I_j|\boldsymbol{c} = c_i)}. \tag{11}$$

Separating P and Q from the above equation by logarithmic transformation, we have

$$\min_{\theta, c_i, s_i} \sum_{i=1}^{N} \sum_{I_j \sim P_i} P_i(\boldsymbol{x} = I_j|\boldsymbol{c} = c_i) \log P_i(\boldsymbol{x} = I_j|\boldsymbol{c} = c_i)$$
$$- P_i(\boldsymbol{x} = I_j|\boldsymbol{c} = c_i) \log Q_{\theta, s_i}(\boldsymbol{x} = I_j|\boldsymbol{c} = c_i). \tag{12}$$

Since $P_i(\boldsymbol{x} = I_j|\boldsymbol{c} = c_i)$ is the dataset distribution, which is an unknown constant distribution, therefore, the first term is a constant, the optimization target is equivalent to

$$\max_{\theta, c_i, s_i} \sum_{i=1}^{N} \sum_{I_j \sim P_i} P_i(\boldsymbol{x} = I_j|\boldsymbol{c} = c_i) \log Q_{\theta, s_i}(\boldsymbol{x} = I_j|\boldsymbol{c} = c_i). \tag{13}$$

Rewriting it into mathematical expectation form, we have

$$\max_{\theta, c_i, s_i} \sum_{i=1}^{N} \mathbb{E}_{I_j \sim P_i} \log Q_{\theta, s_i}(\boldsymbol{x} = I_j|\boldsymbol{c} = c_i), \tag{14}$$

where $P_i$ refers to $P_i(\boldsymbol{x} = I_j|\boldsymbol{c} = c_i)$. Our optimization target is equivalent to maximum likelihood estimation. Here we assume Q is a Gaussian distribution,

$$Q_{\theta, s_i}(\boldsymbol{x}|\boldsymbol{c} = c_i) = \frac{1}{\sqrt{2\pi}\sigma} \exp\left(-\frac{1}{2\sigma^2} \|x - G_\theta(s_i, c_i)\|_2^2\right). \tag{15}$$

Combining Eq. 14 and Eq. 15, we have

$$\max_{\theta, c_i, s_i} \sum_{i=1}^{N} \left(-\frac{1}{2\sigma^2} \|I_i - G_\theta(s_i, c_i)\|_2^2\right). \tag{16}$$

Consequently, the final optimization target is

$$\min_{\theta, c_i, s_i} \sum_{i=1}^{N} \|I_i - G_\theta(z_i)\|_2^2. \tag{17}$$

Q.E.D.

