# OpenReview forum: "Rethinking Content and Style: Exploring Bias for Unsupervised Disentanglement"
_ICLR.cc/2021/Conference — Reject_

### Official Review · AnonReviewer4 · 2020-10-27
**Interesting results but possible limited novelty**

**Rating:** 7
**Confidence:** 4

**Review:**

The paper addresses the problem of unsupervised content style disentanglement. To this end, a content vector and a style vector is sampled from a given prior distribution. The style vector is decomposed to mean and std which is applied on the content code in the same manner as in AdaIN (C-S Fusion block).

Pros:

The paper is well written and clear, providing a clear formulation of the problem and the proposed architecture.

The method provided is well motivated and clear.

The experiments demonstrate some improvement on state of the art. Some ablation is performed on the use of normalization layer, showing its effect on disentanglement. A variety of datasets are considered, and both generation quality (LPIPS) and disentanglement (classification accuracy) are numerically evaluated. Overall the experimental evaluation is extensive and show some improvement over a number of baselines as well as a new application in 3D generation.

Cons:

To me the C-S fusion block is essentially the application of AdaIn from the style vector to the content vector (hence applying a change of style). Modeling the content vector as a shared parameter c and the style vector as an image-specific has been introduced in MUNIT before (and in other works). Other than architectural specific choices, the difference to MUNIT is the application of this framework in conjunction with GLO instead of using and style encoder and a content encoder. Even though MUNIT was designed to work with class level supervision, the method suggested seems very similar (other then the use of GLO instead of encoders) and it might be the case that MUNIT would therefore perform similarly. I therefore believe a comparison to MUNIT when both A and B are the same dataset (e.g celebA) would provide a better insight.

Image2StyleGAN [1] and StyleGANv2 [2] are also able to disentangle content and style. It would be interesting to compare their method to the one suggested. The very latest work tackling this problem is that of Swapping Autoencoders [3] but this has only been recently published.

Some concerns regarding the experiments: In table 2, the results of disentanglement are worse that that FactorVAE. Why is this the case? In addition, no comparison to baselines is provided for the Figure 7 - visual analogies on the Market-1501 dataset.

[1] Image2stylegan: How to embed images into the stylegan latent space? In: IEEE International Conference on Computer Vision (ICCV) (2019)

[2] Analyzing and Improving the Image Quality of StyleGAN. Karras et al.,

[3] Swapping Autoencoder for Deep Image Manipulation. NeurIPS 2020.

---

> ### Author Response · Authors · 2020-11-22
> **More experimental results and clarifying concerns**
>
> Thanks for your comments and suggestions. We provide the required comparison in the revised submission and answer all the concerns below.
>
> - Claim of novelty and AdaIN.
>
> Starting with our assumption that content embeddings share the same distribution, and leveraging AdaIN-like operation, we achieve unsupervised content and style disentanglement without needing “swapping” operation and GAN loss constraint to extract the shared content information as image translation works (MUNIT [1] and Park et al.[2]) do.  We also add this part in our revised submission.
>
> AdaIN serves different purposes between MUNIT [1] and our work.
>
> Our goal is to use AdaIN-like operation to force the content embeddings to follow a shared distribution. Please note that AdaIN is only one special case of our proposed C-S fusion block when the shared distribution is chosen to be N(0, I).  There will be different operations according to the different forms of the shared distribution, e.g., we use L2 normalization to approximate uniform distribution. Our method can not work at all without our proposed C-S fusion block, as Figure 2 in our paper shows. However, AdaIN is not a necessary component for MUNIT. Recently, Liu et al.[4] demonstrates in the table below that MUNIT can still work without AdaIN: “Replacing AdaIN with simple concatenation does not affect the level of C-S disentanglement”.  Moreover, As Figure 10 in the MUNIT paper shows, AdaIN can not work alone.
>
> ======================================
>
> Metric&nbsp;&nbsp; &nbsp;&nbsp;&nbsp;&nbsp;&nbsp;&nbsp;|&nbsp;&nbsp; &nbsp;&nbsp;MUNIT&nbsp;&nbsp;&nbsp;|&nbsp; &nbsp;MUNIT w/o AdaIN
>
>
> DC(C, S) (↓)   | 0.44 ±0.06      |     0.43 ±0.01
>
> DC(I, C) (↑)    | 0.57 ±0.07      |      0.58 ±0.08
>
> DC(I, S) (↑)    | 0.70 ±0.02      |      0.56 ±0.03
>
> IOB(I, C) (↑)   | 4.36 ±0.38      |      4.85 ±0.10
>
> IOB(I, S) (↑)   | 1.31 ±0.04      |      1.17 ±0.04
>
> ======================================
>
> DC (C,S) is a metric for C-S disentanglement and lower is better.
>
>  - Comparison to MUNIT [1] and Park et al. [2].
>
> We follow your suggestion to compare our method to MUNIT [1] and Park et al. [2]. The result is shown in Figure 14 in the appendix, and we provide analysis in Appendix F.
>
> - Comparison to StyleGAN2 [3].
>
> In our framework, the optimized content (convolution branch) and style embeddings are disentangled representations of corresponding images. While StyleGAN2 [3] keeps the input of the convolution branch as a learnt constant for the whole dataset and finds the feature space of the “style” branch has disentanglement ability. The results of styleGAN2 [3] are shown in Figure 15 in the appendix.
>
> - In Table 2, the results of disentanglement are worse that that FactorVAE.
>
> The size of the dimension of embeddings in FactorVAE is ten. While in our method, we set the size of style embeddings to 256 and content embeddings to 128. As we claimed in section 4.1,  “For FactorVAE (Kim & Mnih, 2018), the poor reconstruction quality indicates that the latent embeddings encode a very small amount of information that can hardly be classified. The dimensions of the latent vectors of different methods vary from ten to hundreds. Actually, the higher dimension usually leads to easier classification. Based on the above observations, the classification metric may not be appropriate for disentanglement, which is also observed in Liu et al. (2020).” Therefore, the lower classification results of FactorVAE are due to the low dimension and low information capacity of the embeddings.
>
> - Visual analogies on the Market-1501 dataset.
>
> We follow your suggestion and show qualitative comparison with baselines on ReID dataset Market-1501, as shown in Figure 19 in the appendix.
>
> [1] Xun Huang, Ming-Yu Liu, Serge Belongie, and Jan Kautz. Multimodal unsupervised image-to-image translation. In ECCV, 2018.
>
> [2] Taesung Park, Jun-Yan Zhu, Oliver Wang, Jingwan Lu, Eli Shechtman, Alexei A. Efros, Richard Zhang. Swapping Autoencoder for Deep Image Manipulation. In NeurPIS, 2020.
>
> [3] Analyzing and Improving the Image Quality of StyleGAN. Tero Karras, Samuli Laine, Miika Aittala, Janne Hellsten, Jaakko Lehtinen, Timo Aila. In CVPR, 2020.
>
> [4] Metrics for Exposing the Biases of Content-Style Disentanglement. Xiao Liu, Spyridon Thermos, Gabriele Valvano, Agisilaos Chartsias, Alison O'Neil, Sotirios A. Tsaftaris. In arxiv.

---

### Official Review · AnonReviewer1 · 2020-10-27
**Nice experimental results, but lack of thorough investigation and formalism limit impact**

**Rating:** 4
**Confidence:** 4

**Review:**

Summary: This paper presents a novel combination of losses to learn representations that disentangle content from style. An ablation study is done with a few variants of the model, and extensive quantitative and qualitative experimental results on content/style disentangling demonstrate the model’s performance.

Overall the experimental results look very nice and seem to compare well to other methods both  qualitatively and quantitatively. However, the scope of the contribution seems limited since the losses have already been introduced in the literature. The novelty seems to be in the embedding combination method, which is not explored as thoroughly as I’d expect. Also, the grounding of the proposed loss as a probabilistic model seems lacking to me.

Strengths:
* Impressive looking results on style transfer in a few distinct domains, including faces (celeba), chairs, and Car3D.
* Simple method that could in principle be applied to a variety of domains.

Weaknesses:
* This paper would be stronger if it dispensed with the incorrect formalism introduced in Section 3, and introduced the training objective as a set of standard loss functions as in Eq. 4.
* If this paper is about methods for combining embeddings for content/style recombination, I would hope to see a more thorough exploration of the different options. For example,  Eq. 2 is close to a bilinear model, as used in [4]. It would be interesting to see this combination method compared to explicitly, as it is a well known method for linear content/style recombination. It seems like only two methods (concatenation and the proposed method) were tried.
* The proposed method seems to be an ad-hoc combination of losses already introduced in the literature.
* “most previous C-S disentanglement works rely on supervision, which is hard to obtain for real data” --- full supervision of both content and style is difficult to obtain, but supervision of one variable alone is extremely common and represents the basis for most modern content/style decomposition methods. For example, see generic content/style decomposition models based on GANs [1, 3], and VAEs [2].
* Denton & Birodkar 2017 does not use full supervision of content and style, rather they assume that the static components in the video sequence represent content, and everything else represents style.
* Page 1 Paragraph 3: This definition of content --- as  factors shared across the whole dataset --- doesn’t make sense to me. Each face image has both an identity and a pose, so why is one content and not the other? It seems to me that the distinction is somewhat arbitrary, and probably determined by a target downstream task (e.g., face classification).

Clarity:
* Section 3.1: This needs a more complete explanation. A few detailed questions follow:
** Section 3.1: Is Q conditioned on s as well as c?
** Equation 1: Since P(x|c) is NOT conditioned on s, wouldn’t this objective function encourage Q to be independent of s, i.e. Q(x|c,s) = Q(x | c) ?
** Where is \Psi in Eq. 1 ?
* Eq. 3 - I don’t follow how this is optimizing KL( P(x | c) || Q_{s, \theta}(x | c ) ). Please give a derivation.
* Eq. 2+3 s_i, c_i are free parameters that the network is allowed to optimize, so how are  images not in the training set dealt with? Does it require training a new s_i, c_i for each new sample?
* Which loss is Eq. 3 part of? I don’t see it listed in Eq. 4.

Rating:
References:
[1] Xun Huang,  Ming-Yu Liu,  Serge Belongie,  and Jan Kautz.   Multimodal unsupervised image-to-image translation. InECCV, 2018.
[2] Diederik P Kingma, Shakir Mohamed, Danilo Jimenez Rezende, and Max Welling. Semi-supervisedlearning with deep generative models.  InAdvances in Neural Information Processing Systems,pp. 3581–3589, 2014
[3] Michael F Mathieu, Junbo Jake Zhao, Junbo Zhao, Aditya Ramesh, Pablo Sprechmann, and377Yann LeCun. Disentangling factors of variation in deep representation using adversarial train-378ing. InAdvances in Neural Information Processing Systems, pp. 5040–5048, 2016
[4] Tenenbaum, J. B., & Freeman, W. T. (2000). Separating style and content with bilinear models. Neural computation, 12(6), 1247-1283.

---

> ### Author Response · Authors · 2020-11-22
> **Revised formulation and clarifying concerns**
>
> Thanks for your comments and suggestions. We provide a revised submission and answer all the concerns below.
>
> - Formulation and objective.
>
> Thanks for your suggestion. We provide proof in Appendix H, and we also revise the formulation of Section 3 of our revised submission.
>
> - Contribution and  more thorough exploration of the different options  for combining embeddings for content/style recombination.
>
> This combination operation of content and style is not our contribution. Starting with our assumption that content embeddings share the same distribution and leveraging AdaIN-like operation, we achieve unsupervised content and style disentanglement. We also tried a bilinear model and found it did not have disentanglement ability in our case.
>
> - The proposed method seems to be an ad-hoc combination of losses already introduced in the literature.
>
> Perceptual loss serves as the reconstruction loss, which is the same with [1,2]. The instance discrimination loss and information bottleneck loss are auxiliary to help disentanglement. And we are the first to propose to use instance discrimination in the disentanglement area.
>
> - Supervision.
>
> The supervision of some specific domain images is not well available. For example, it is tough to collect videos of paintings and cartoons. Indeed, it is possible to obtain supervision of one variable alone, e.g., images from different domains or video images, but it is easier to collect images with less constraint, e.g., existing web images.
>
> - Definition of content.
>
> Conceptually, the identity can also be defined as the shared one, but in our setting, we take the pose of face images as the shared one rather than the identity. The reasons are two folds. First, as the reviewer mentioned, it is determined by downstream tasks, such as 3D reconstruction, face frontalization and novel view synthesis. Second, it is dependent on the dataset. For CelebA, the images with different identities but the same pose is much more common than images with the same identity but different poses.
>
> - Section 3.1: Is Q conditioned on s as well as c?
>
> Q is conditioned on c but parameterized by s.
>
> - Equation 1: Since P(x|c) is NOT conditioned on s, wouldn’t this objective function encourage Q to be independent of s, i.e. Q(x|c,s) = Q(x | c) ?
>
> Please note that Pi(x|c) is the distribution to model Ii, and we  reparameterized Q with si as Qs_i to approximate Pi.
>
> - Where is \Psi in Eq. 1 ?Where is \Psi in Eq. 1 ?
>
> We formulate the constraint that the content embeddings follow the shared distribution as an explicit constraint on \Psi item to Eq1 in our revised submission.
>
> - Optimizing KL( P(x | c) || Q_{s, \theta}(x | c ) ).
>
> Optimizing Eq3 is equivalent to optimize reconstruction loss, which is proved in Appendix H.
>
> - How are images not in the training set dealt with?
>
> We provided the details for unseen images inference in Appendix and we move it to the main paper in the revised submission now.
>
> - Which loss is Eq. 3 part of? I don’t see it listed in Eq. 4.
>
> We optimize reconstruction loss to optimize Eq3, which is proved in Appendix H.  Perceptual loss serves as the reconstruction loss, which is the same with [1,2].
>
> [1] Shangzhe Wu, Christian Rupprecht, Andrea Vedaldi.  Unsupervised Learning of Probably Symmetric Deformable 3D Objects from Images in the Wild. In CVPR 2020.
>
> [2] Aviv Gabbay, Yedid Hoshen. Demystifying Inter-Class Disentanglement. In ICLR 2020.

---

### Official Review · AnonReviewer2 · 2020-10-29
**Unsupervised Content-Style Disentanglement for Image translation**

**Rating:** 4
**Confidence:** 4

**Review:**

In this paper, authors introduce a new approach to content-style (C-S) disentanglement for multimodal unsupervised image-to-image translation. The main idea behind the proposed method is that the content information is encoded into the latent space common for both source and target domains, while the domain-specific style information is used to shift and scale the points from content distribution so they are mapped by the generator to the target domain distribution.

There are a few issues in this paper that I would like to be addressed.

1. In the first page of the paper you write: "When group observation is not available, we define content includes the factors shared across the whole dataset, such as pose.  Take the human face dataset CelebA (Liu et al., 2015) as an example, the content encodes pose, and style encodes identity, and multi-views of the same identity have the same style embeddings, but different content embeddings, i.e., poses". In this paper, authors view only pose information of the faces in CelebA dataset as content; but according to the definition above, shouldn't facial expression also be included in the content as shared information across all examples?

2. The proposed method suggests using style embedding to shift the content embedding before translation in Single case and within translation in Multiple case. How to address the fact that the same idea was used in a few other papers for multimodal cross-domain translation, such as MUNIT by Huang et al. ECCV'2018 or FUNIT by Liu et al. ICCV'2019?

3. In addition to the main C-S fusion block, two more losses were introduced in this method: Instance Discrimination (ID) and Information Bottleneck (IB). To see the effect of each component, it would be helpful to see the ablation study results.

On the other hand, the paper is well-written and well-structured ans easy to follow; the translation results look promising. In addition, the quantitative metrics used in this paper, in particular the content transfer metric, are very reasonable for evaluation of disentanglement quality.

---

> ### Author Response · Authors · 2020-11-22
> **Response to Reviewer 2**
>
> Thanks for your comments. We would like to answer all the concerns.
>
> - Shouldn't facial expression also be included in the content as shared information across all examples.
>
> When we increase the information capacity of content space, e.g., we change the dimension of content embeddings and style embeddings, the facial expression information is disentangled into content space, as shown in  Figure 18 in the appendix.  We choose to disentangle pose, which is more useful for downstream tasks, like 3D reconstruction, face frontalization and novel view synthesis.
>
> - How to address the fact that the same idea was used in a few other papers for multimodal cross-domain translation.
>
> Starting with our assumption that content embeddings share the same distribution and leveraging AdaIN-like operation, we achieve unsupervised content and style disentanglement without needing “swapping” operation and GAN loss constraint to extract the shared content information as MUNIT do. We also add this part in our revised submission.
>
> The motivation for shifting and scaling is inspired by the style transfer works, such as MUNIT [1].
>
> However, AdaIN serves different purposes between MUNIT and our work.
> Our goal is to use AdaIN-like operation to force the content embeddings to follow a shared distribution. Please note that AdaIN is only one special case of our proposed C-S fusion block when the shared distribution is chosen to be N(0, I). There will be different operations according to the different forms of the shared distribution, e.g., we use L2 normalization to approximate uniform distribution. Our method can not work at all without our proposed C-S fusion block, as Figure 2 in our paper shows. However, AdaIN is not a necessary component for MUNIT. Recently, Liu et al.[4] demonstrates in the table below that MUNIT can still work without AdaIN: “Replacing AdaIN with simple concatenation does not affect the level of C-S disentanglement”. Moreover, As Figure 10 in the MUNIT paper shows, AdaIN can not work alone.
>
> ======================================
>
> Metric&nbsp;&nbsp; &nbsp;&nbsp;&nbsp;&nbsp;&nbsp;&nbsp;|&nbsp;&nbsp; &nbsp;&nbsp;MUNIT&nbsp;&nbsp;&nbsp;|&nbsp; &nbsp;MUNIT w/o AdaIN
>
>
> DC(C, S) (↓)   | 0.44 ±0.06      |     0.43 ±0.01
>
> DC(I, C) (↑)    | 0.57 ±0.07      |      0.58 ±0.08
>
> DC(I, S) (↑)    | 0.70 ±0.02      |      0.56 ±0.03
>
> IOB(I, C) (↑)   | 4.36 ±0.38      |      4.85 ±0.10
>
> IOB(I, S) (↑)   | 1.31 ±0.04      |      1.17 ±0.04
>
> ======================================
>
> DC (C,S) is a metric for C-S disentanglement and lower is better.
>
> - Ablation study.
>
> We have provided the ablation study of  Instance Discrimination (ID) and Information Bottleneck (IB), presented in Appendix E.
>
> [1] Xun Huang, Ming-Yu Liu, Serge Belongie, and Jan Kautz. Multimodal unsupervised image-to-image translation. In ECCV, 2018.
>
> [2] Metrics for Exposing the Biases of Content-Style Disentanglement. Xiao Liu, Spyridon Thermos, Gabriele Valvano, Agisilaos Chartsias, Alison O'Neil, Sotirios A. Tsaftaris. In arxiv.

---

### Author Response · Authors · 2020-11-25
**Revision of Submission**

We thank all the reviewers for providing us with valuable feedback. We have modified the main paper as well as the appendix. Hope the revision could address the concerns from the reviewers. Here are the major changes that we made to the manuscripts. All modifications are highlighted by the blue color in the revised submission.

In the main paper:

1. We have revised the problem formulation in **Section 3.1** and **3.2**.

2. We have added a visualization of the training procedure and related analysis in **Section 3.3**.

3. We have added a comparison between image translation works in **Section 4.2**.

4. We have moved the inference for unseen image from appendix to **Section 4.3**.

5. We have moved the result on ReID dataset and added comparison with baselines to **Appendix D.2**.

In the Appendix:

1. We have added **Appendix F** to compare with StyleGAN2 and image translation works.

2. We have added **Figure 18** to study on the influence of size of embeddings.

3. We have added a proof in **Appendix H** on our formulation.

**Clarification of novelty**: We provide a interesting insight for unsupervised C-S disentanglement. Starting with our assumption that content embeddings share the same distribution, and leveraging AdaIN-like operation, we achieve unsupervised content and style disentanglement by forcing the content embeddings to follow a shared distribution. Please note that AdaIN is only one special case of our proposed C-S fusion block when the shared distribution is chosen to be N(0, I). Moreover, AdaIN-like operation serves different purposes between image translation works and our work.

---

### Decision · Program_Chairs · 2021-01-07
**Final Decision**

**Decision:**

Reject

**Comment:**

The paper proposes an approach to defining/tackling the question of separating "style" and "content" of images, and introduces a novel way to learn representation that disentangle these aspects of images. I think it offers some new ideas. The reviewers were split on the evaluation. Among the chief concerns with the initial submission were a problematic formulation of the objective, missing comparisons and analysis, and questions about novelty of the architecture (in particular w.r.t. AdaIn). I think the rebuttal/revision have addressed these fairly well. I do agree with R2 that some flaws remain, in particular the analysis could be more thorough/complete, and the paper could then be stronger.